# Arbitrary engineering of spatial caustics with 3D-printed metasurfaces

Xiaoyan Zhou [1,2,3], Hongtao Wang [2,3] ✉, Shuxi Liu[1], Hao Wang [3], John You En Chan[3], Cheng-Feng Pan [2,3], Daomu Zhao [1] ✉, Joel K. W. Yang [3] ✉ & Cheng-Wei Qiu [2] ✉

Caustics occur in diverse physical systems, spanning the nano-scale in electron microscopy to astronomical-scale in gravitational lensing. As envelopes of rays, optical caustics result in sharp edges or extended networks. Caustics in structured light, characterized by complex-amplitude distributions, have innovated numerous applications including particle manipulation, high-resolution imaging techniques, and optical communication. However, these applications have encountered limitations due to a major challenge in engineering caustic fields with customizable propagation trajectories and in-plane intensity profiles. Here, we introduce the "compensation phase" via 3D-printed metasurfaces to shape caustic fields with curved trajectories in free space. The in-plane caustic patterns can be preserved or morphed from one structure to another during propagation. Large-scale fabrication of these metasurfaces is enabled by the fast-prototyping and cost-effective two-photon polymerization lithography. Our optical elements with the ultra-thin profile and sub-millimeter extension offer a compact solution to generating caustic structured light for beam shaping, high-resolution microscopy, and light-matter-interaction studies.

Caustics represent ubiquitous phenomena in various wave domains, including light waves, rogue waves, and gravitational waves[1,2]. In optics, caustics manifest as bright patches or edges, resulting from the interaction of light with smoothly curved surfaces. These phenomena can be observed as bright focal curves refracted by common objects like a glass cup, intricate light networks beneath glass surfaces, or under the surfaces of shallow bodies of water due to the wavy structures on their surfaces[1,2]. From a physical perspective, an optical caustic is the tangent curve or surface formed by light rays, delineating the envelope of concentrated light rays, a curve that encapsulates the highest intensity regions[3]. The classification of caustics is based on their geometric and topological characteristics, resulting in seven elementary catastrophes[4–6]. For example, caustic Airy beams are associated with fold catastrophes[7,8], Pearcey beams correspond to cusp catastrophes[9–11], and Swallowtail beams represent swallowtail catastrophes[12]. Such caustics have found advanced applications in optical trapping[13,14], material processing[15,16], high-resolution microscopy[17,18], and communication technology[19]. Nevertheless, these applications face limitations due to a major hurdle in engineering caustic fields with customizable propagation trajectories and in-plane intensity profiles. A leap forward in the multi-dimensional customization of caustics will unlock considerable benefits. For example, these artificial caustic beams can strategically circumvent obstacles, providing an advanced mechanism for the 3D manipulation of particles[13,14]. This innovation also holds promise in optical communications, where it can minimize signal loss and enhance system stability[19]. Additionally, caustic light with precisely defined curvilinear borders facilitates customized high-energy transfer, which is

[1]Zhejiang Key Laboratory of Micro-nano Quantum Chips and Quantum Control, School of Physics, Zhejiang University, Hangzhou 310058, China. [2]Department of Electrical and Computer Engineering, National University of Singapore, Singapore 117583, Singapore. [3]Engineering Product Development, Singapore University of Technology and Design, Singapore 487372, Singapore. ✉e-mail: hongtao_wang@sutd.edu.sg; dmz123@zju.edu.cn; joel_yang@sutd.edu.sg; chengwei_qiu@nus.edu.sg

particularly beneficial in nano-fabrication[15,16]. Evolving from natural occurrences to structured light, and now advancing towards advanced engineering, optical caustics are poised for significant breakthroughs, not only in theoretical research and experimental techniques, but also in practical applications (Supplementary Note 1). While the shape of a caustic is a function of the optical system, the distinguishing features of the caustic are consistent across all systems, *i.e.*, within the standard families of caustics[20–22]. Thus, achieving arbitrary engineering of spatial caustics remains challenging, despite successful demonstrations of customizing caustics into structured light in a single dimension[23].

Benefiting from the advances in nanofabrication technology, including electron beam lithography (EBL), focused ion beam (FIB) lithography, and two-photon polymerization lithography (TPL), sub-wavelength nanostructures enable simultaneous control of both the amplitude and phase of incident light waves[24–33]. Compared to controlling only amplitude or phase, the complex-amplitude modulation of incident light gives more precise manipulation of light fields[24,25,30,32]. This modulation is vital for the realization of optical caustics, which are structured in amplitude and phase[1,2,4–6]. Additionally, optical meta-atoms with wavelength-scale dimensions provide quasi-continuous spatial control of light while avoiding higher-order diffraction channels[34–39]. The ability of metasurfaces to manipulate light in tighter spaces can lead to more potential applications in caustic engineering[40]. Among them, 3D-printed metasurfaces stand out due to their advantages such as less complex post-lithography processes, large-scale fabrication capability, and the ability to sculpt complex 3D structures[41–48]. One type of meta-atoms is 3D-printed nanofins[32,46,47]. These structures function as truncated waveguides with sub-micrometer lateral dimensions and several micrometers height. They offer an additional degree of freedom in height control, resulting in larger real-time delays and a broader working bandwidth compared to traditional plasmonic or dielectric nanoantenna[48]. With this unleashed height degree of freedom, anisotropic nanofins offer an extensive 3D meta-atom library, allowing the independent and complete polarization, phase or amplitude modulation[32,46,47]. These polymeric meta-surfaces are interfaced with fiber end-faces to perform the dispersion engineering of fiber-optic devices[46] or transform the output into diverse structured light[47]. In an optical configuration involving crossed circular polarization, the phase of the incident beam is spatially modulated based on the Pancharatnam–Berry (PB) phase, whereas the amplitude is independently controlled by adjusting the height of the nanofins[32]. This unique combination of features makes complex-amplitude 3D-printed metasurfaces an ideal platform for efficient and precise engineering of spatial caustics.

Here, we use a 3D-printed metasurface to sculpt arbitrary caustic patterns with anticipated propagation trajectories in free space (Fig. 1a). The metasurface is encoded with complex-amplitude information obtained by performing a Fourier transform (FT) of the amplitude and phase distributions of caustic beams in Fourier space (Fig. 1b(i)–d(i)). This metasurface enables the lensless reconstruction of versatile caustic structured light fields (Fig. 1b–d). In our approach, the desired caustic beam can also propagate along curved trajectories (Fig. 1c(ii)) instead of merely linear trajectories (Fig. 1b(ii)). Furthermore, the caustics can be designed to exhibit multiple intensity distributions over varying propagation distances (Fig. 1d(ii)).

## Results

### Constructing caustics into a spatial focal curve

We consider a single caustic point that propagates along a curved trajectory in 3D space (Fig. 2a–c). This trajectory produces a spatial focal curve that is fundamental for arbitrary caustic engineering.

To realize this spatial focal curve, we design the initial amplitude and phase distributions in Fourier space and then calculate the light field by using the method of angular spectrum[34]. The curve is decomposed into two analytical functions given by $X(z)$ and $Y(z)$ describing the trajectories of the beams in the $x$–$z$ and $y$–$z$ planes respectively, where $z$ is the coordinate on the optical axis. The angular spectrum integral is then asymptotically calculated using the method of stationary phase[3–6]. Thus, the solutions $(k_x, k_y)$ form a continuous locus in the spatial frequency domain, which is expressed explicitly as a circle $(k_x - kX'(\tilde{z}))^2 + (k_y - kY'(\tilde{z}))^2 = \beta^2(\tilde{z})$, where $(k_x, k_y)$ refers to the wavevector; $k = 2\pi/\lambda$ is the wavenumber with $\lambda$ being the wavelength; $X'(\tilde{z})$ and $Y'(\tilde{z})$ are the first-order derivatives of $X(\tilde{z})$ and $Y(\tilde{z})$ respectively; $\beta(\tilde{z})$ represents the radius. That is to say, the wavevectors of the light rays that form a caustic point on this focus curve constitute a circle in the Fourier space. If any wavevector $(k_x, k_y)$ from the locus is

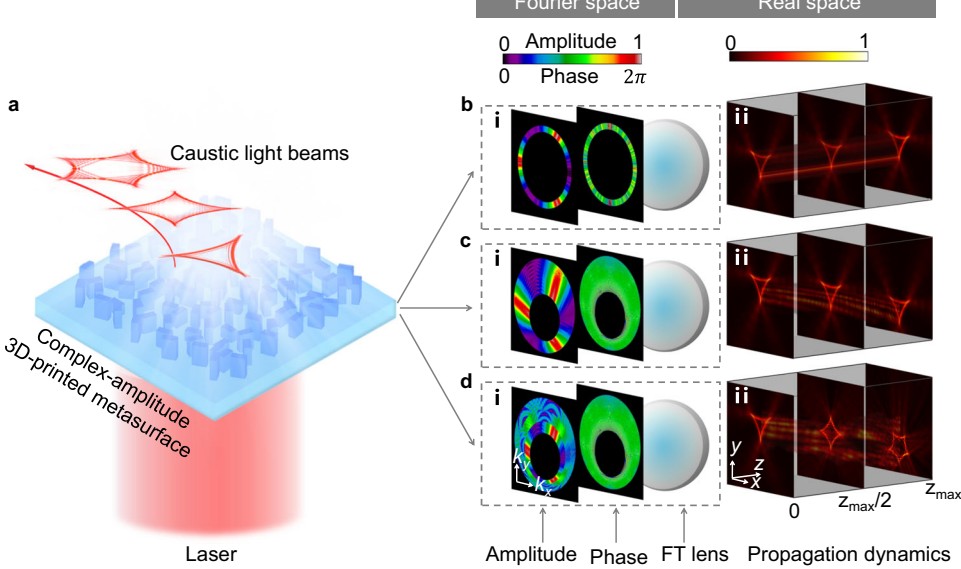

**Fig. 1 | Principle of arbitrary engineering of spatial caustics based on 3D-printed metasurfaces. a** Schematic of customized optical caustics, generated by the metasurface. **b(i)–d(i)** Amplitude and phase information of the desired caustic beams in Fourier space, together with a FT lens. **b(ii)–d(ii)** Propagation dynamics of realized caustic structured light in real space.

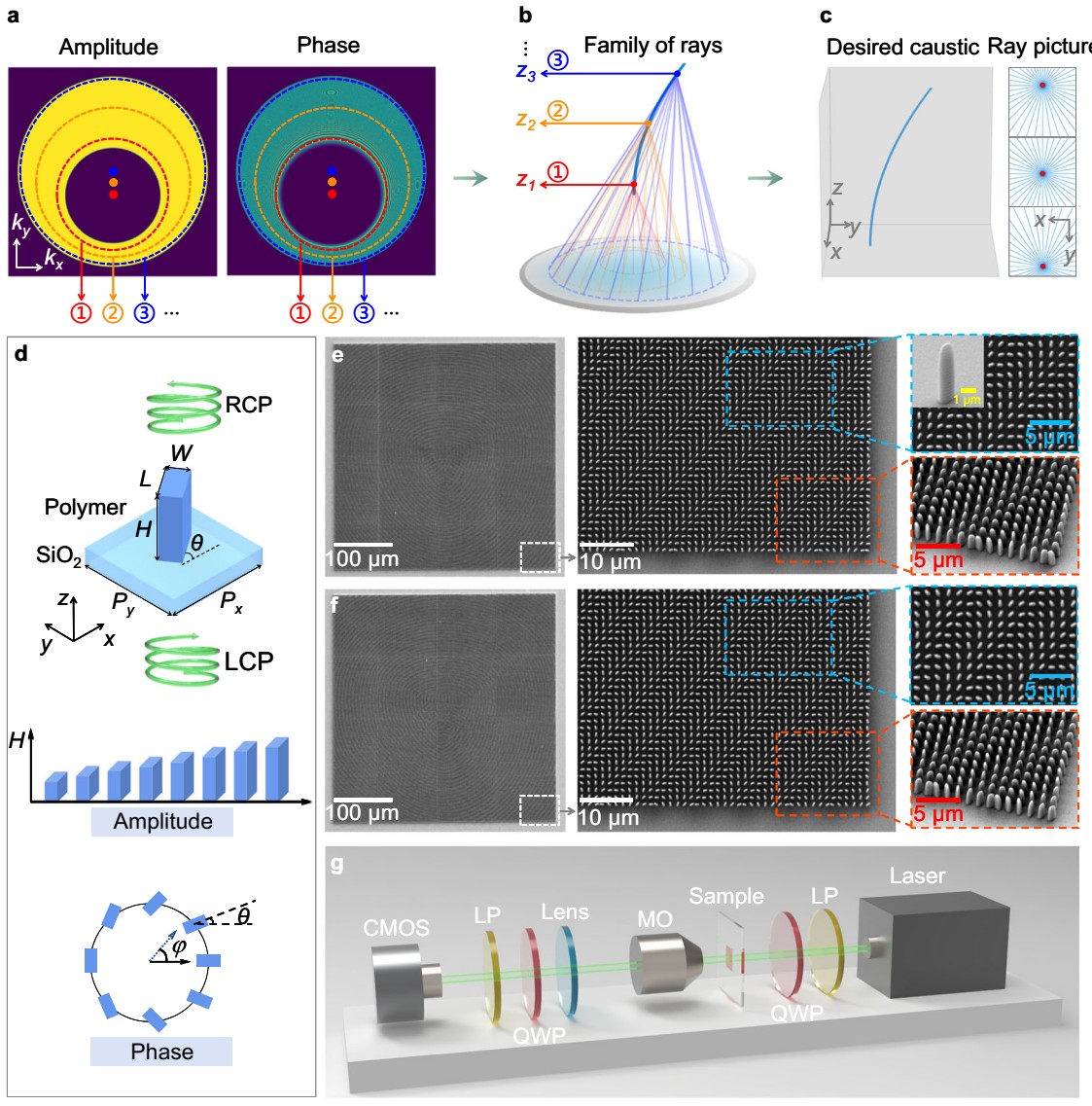

**Fig. 2 | The design principle of arbitrary spatial caustic engineering, the fabrication of complex-amplitude 3D-printed metasurfaces, and the characterization of optical caustics. a** Initial complex-amplitude distributions of the spatial focal curve in Fourier space. The dots are the circle centers corresponding to different propagation distances. **b** Rays emitted from the circles intersect on a specified focal curve. **c** Visualization of the caustic curve and transverse projections of rays (blue) aligning with the caustics (red) at $z = 0$, $z = z_{max}/2$, and $z = z_{max}$, respectively. **d** Schematic of one meta-atom. The parameters $W$, $L$, $H$ and $\theta$ denote the width, length, height and in-plane rotation angle of the polymer nanofin,

respectively. LCP and RCP refer to left- and right-handed circular polarized light. The height ($H$) and in-plane rotation ($\theta$) enable independent control of both the amplitude and phase characteristics of the transmitted light. **e, f** SEM images of the metasurfaces for cases in Fig. 1c, d, together with their magnified images in the top- and oblique views, as well as an image of a single element. **g** Schematic of the experimental setup for optical caustic characterization. LP linear polarizer, QWP quarter-wave plate, MO microscope objective; CMOS complementary metal oxide semiconductor.

mapped to the distance $\tilde{z}$, a two-variable function $\tilde{z}(k_x, k_y)$ is obtained. From this perspective, the movement of the circle center $(kX'(\tilde{z}), kY'(\tilde{z}))$ serves as the fundamental physical mechanism in driving the curved propagation trajectory. To achieve a proper design of spatial focal curve, the displacement of the circle center in Fourier space should be no more than the variation of radius $\beta(\tilde{z})$. Finally, we obtain the phase in spatial frequency domain with normalized unity amplitude:

$$\Phi(k_x, k_y) = \frac{k}{2} \int_0^{\tilde{z}} \left[ \left( \frac{dX(\xi)}{d\xi} \right)^2 + \left( \frac{dY(\xi)}{d\xi} \right)^2 - \left( \frac{\beta(\xi)}{k} \right)^2 \right] d\xi - k_x X(\tilde{z})$$
$$- k_y Y(\tilde{z}) + \frac{k_x^2 + k_y^2}{2k} \tilde{z}. \tag{1}$$

Note that $0 \le \beta(\tilde{z}) \le k$, which limits the maximum propagation distance $z_{max}$ of the caustic point. More details of the derivation are provided in Supplementary Note 2.

We present an example of the spatial focal curve with a parabolic trajectory denoted as $X(z) = 0$, $Y(z) = 2 \times 10^{-7} z^2/\lambda$, lying on the plane $x = 0$. The maximum propagation distance is set as $z_{max} = 3 \times 10^4 \lambda$ in the far field. The calculated amplitude and phase distributions in Fourier space are shown in Fig. 2a (see Supplementary Note 3 for detailed phase information). We use red (①), orange (②), and blue (③) dashed circles to represent the complex-amplitude distributions corresponding to the propagation distances $z_1 = 0.1 z_{max}$, $z_2 = 0.5 z_{max}$, and $z_3 = 0.9 z_{max}$, respectively (Fig. 2a, b). The red, orange, and blue solid dots in Fig. 2a denote the positions of the centers of these circles. As the propagation distance increases from $z_1$ to $z_3$, the center of the circle moves along the $k_y$ axis, which is correctly predicted by the previous

physical analysis. The modulated optical field then passes through a converging lens (at the bottom of Fig. 2b). Thus, the conical ray bundles emitted from the circles intersect at corresponding points on the curve, which form the caustic points that propagate along the parabolic trajectory. Figure 2b illustrates the formation of this parabolic focal curve. To investigate the caustic characteristics of this curve, we simulated the transverse projections of optical ray (blue) and caustics (red) at $z = 0$, $z = z_{max}/2$, and $z = z_{max}$ in Fig. 2c (see Supplementary Note 4 for calculation methods and analysis). The caustic point, which is the singularity of the gradient mapping, shifts in the positive $y$ direction as the propagation distance increases. Moreover, the simulation results of the transverse intensity and phase distributions are provided in Supplementary Note 5. From the transverse profiles at different distances, the intensity maxima of the field, characterized by their caustics, exhibit an accelerated movement along the desired parabolic trajectory under wavefront control.

## Design and characterization of the complex-amplitude 3D-printed metasurface

To achieve compact and lensless caustic engineering, we introduce the design and optimization of the 3D-printed metasurface with complete and independent control of amplitude and phase. The complex-amplitude information encoded in the metasurface is derived from the FT of the angular spectrum. We design the meta-atom to be a rectangular nanofin made of polymerized IP-L photoresist (refractive index ~1.52 in the visible band) (Fig. 2d). This nanofin works as a truncated waveguide with high aspect ratio up to 10. To find the amplitude and initial phase of cross-polarized light transmitted from the nanofin, we use Lumerical finite-difference time domain (FDTD) to simulate the 3D-printed meta-atom with varying height ($H$) and length ($L$), whilst the width-to-length ratio is fixed at 0.5 (see Methods and Supplementary Note 6). Following the principles of the PB phase, we use the in-plane rotation angle ($\theta$) of the nanofin as an additional degree of freedom to tune the phase of cross-polarized light transmitted from the nanofin. The PB phase is given by $\varphi = 2\theta$. Thus, the complex-amplitude of the beam is fully controlled by nanofins with various geometries.

For experimental demonstrations, we choose nanofins with uniform transverse dimensions ($W = 400$ nm and $L = 800$ nm) and varying heights ranging from 3.0 μm to 3.7 μm (represented by the black line in Fig. S4a, b). The characterization results are consistent with the simulation (Fig. S4c). A meta-atom library utilizing both the height and in-plane rotation angle of meta-atoms is created (Fig. S4d). We then fabricate two 3D-printed metasurfaces for the cases in Fig. 1c, d. Each metasurface comprises an array of 300 × 300 nanofins with periodicity 1.25 μm, yielding an area of 375 μm × 375 μm (see Methods and Supplementary Movie for fabrication details). The scanning electron microscopy (SEM) images of the entire metasurfaces are depicted in Fig. 2e, f, respectively. High-magnification and tilted SEM images in the insets show the details of complex-amplitude 3D-printed metasurfaces and the constituent nanofins. For a better assessment of the precision of the nanoprinting, the SEM image of a single nanofin at a 45° observation angle has been inserted. More SEM images of nanofins with different heights (3.0 μm to 5.0 μm) are available in Supplementary Note 7. Even though rounded surfaces can be observed in our fabricated nanofins due to the ellipsoid shape of the diffraction-limited focal voxel in 3D laser printing, the discrepancy in both amplitude and phase modulation between nanopillars with sharp and rounded surfaces is small (Supplementary Note 8). As such, the influence of these rounded surface features on the optical performance is marginal. Through an analysis of the ratio between the laser power diffracted into the desired order and the total incident power on the metasurface, we can determine that the diffraction efficiency of our metasurfaces ranges from 15.41% to 19.59%.

To accurately construct and analyze the caustic structured light field using the 3D-printed metasurfaces, we employ a home-made optical setup as presented in Fig. 2g. First, the emitted light ($\lambda = 532$ nm) from the NKT supercontinuum laser source is converted to a left-circularly polarized (LCP) beam through a linear polarizer and a quarter-wave plate. The beam is then transmitted through the 3D-printed metasurface to obtain the desired complex-amplitude information of the caustic beam. Afterwards, the output is magnified by a microscope objective (Nikon, NA = 0.45, 20×) and a tube lens ($f = 200$ mm), then analyzed by another circular polarizer (composed of a quarter waveplate and a linear polarizer with 45° angle). The higher-order diffraction patterns will not enter the microscope objective due to large diffraction angles (see Supplementary Note 9). Finally, these caustic beams are captured by a complementary metal oxide semi-conductor (CMOS) camera.

## Sculpting caustics into desired patterns propagating along arbitrary trajectories

Here, we assume the caustics of the beam at plane $z$ is a transverse path, which is parametrically expressed as $\mathbf{r}_p(\tau) = [m(\tau), n(\tau)]^T$ with $\tau$ being the length of the path. In this case, the propagation trajectory of any caustic point on the path in the $z$ direction is represented as $(X(z) + m(\tau), Y(z) + n(\tau), z)$. According to Eq. (1), the required phase in Fourier space is

$$\Phi_p(\tau, k_x, k_y) = \frac{k}{2} \int_0^{\tilde{z}} \left[ \left(\frac{dX(\xi)}{d\xi}\right)^2 + \left(\frac{dY(\xi)}{d\xi}\right)^2 - \left(\frac{\beta(\xi)}{k}\right)^2 \right] d\xi - k_x$$
$$\left[X(\tilde{z}) + m(\tau)\right] - k_y \left[Y(\tilde{z}) + n(\tau)\right] + \frac{k_x^2 + k_y^2}{2k} \tilde{z}. \quad (2)$$

The key to designing the light field in real space is to construct the angular spectrum by coherent integration along the desired path $\mathbf{r}_p(\tau)$ with the appropriate amplitude $A(\tau)$ and compensation phase $\phi(\tau)$. The amplitude is chosen as $A(\tau) = 1/\sqrt{|\mathbf{r}'_p(\tau)|} = 1$ to achieve uniform intensity along the path. To ensure that the interference is constructive along the desired path only, the compensation phase would be accumulated through the optical path length $\tau$ by rays forming the desired caustics. The compensation phase (see Supplementary Note 10 for the derivation) is given by

$$\phi(\tau) = \beta(\tilde{z})\tau + \mathbf{G}'(\tilde{z})k[\mathbf{r}_p(\tau) - \overline{\mathbf{r}_p(\tau)}], \quad (3)$$

where $\mathbf{G}'(\tilde{z}) = [X'(\tilde{z}), Y'(\tilde{z})]$ indicates the derivative of the function describing the propagation trajectory; $\mathbf{r}_p(\tau) = [m(\tau), n(\tau)]^T$ represents the transverse path of the caustics; and $\overline{\mathbf{r}_p(\tau)} = \frac{1}{l} \int_0^l \mathbf{r}_p(\tau) d\tau$ signifies the center of the path with $l$ being the total length of the path.

For simplification, we can define $\phi(\tau) = \phi_{length}(\tau) + \phi_{trajectory}(\tau)$. $\phi_{length}(\tau) = \beta(\tilde{z})\tau$ is governed by the length of the transverse path, and $\phi_{trajectory}(\tau) = \mathbf{G}'(\tilde{z})k[\mathbf{r}_p(\tau) - \overline{\mathbf{r}_p(\tau)}]$ is jointly determined by the propagation trajectory and the transverse path of caustics at the corresponding plane $\tilde{z}$. Therefore, the spectrum distribution can be calculated by the following integral:

$$F(k_x, k_y) = \int_0^l \exp\left[i\phi(\tau) + i\Phi_p\left(\tau, k_x, k_y\right)\right] d\tau. \quad (4)$$

As an illustrative example, we present a deltoid-shaped structured light beam with a parabolic propagation trajectory. The trajectory equation and the maximum propagation distance are designed the same as above. For charity, we still choose three specific planes as above, denoted as $z_1$, $z_2$, and $z_3$, to showcase the process of constructing the previous caustic point into a predefined pattern in the transverse plane.

Initially, the focal points on the cross-section are arranged along a deltoid-shaped path, as depicted by the red lines in Fig. 3a. We sequentially superpose these focal points, designated as ❶, ❷, ❸, and so forth, along the transverse path indicated by the arrows in Fig. 3a.

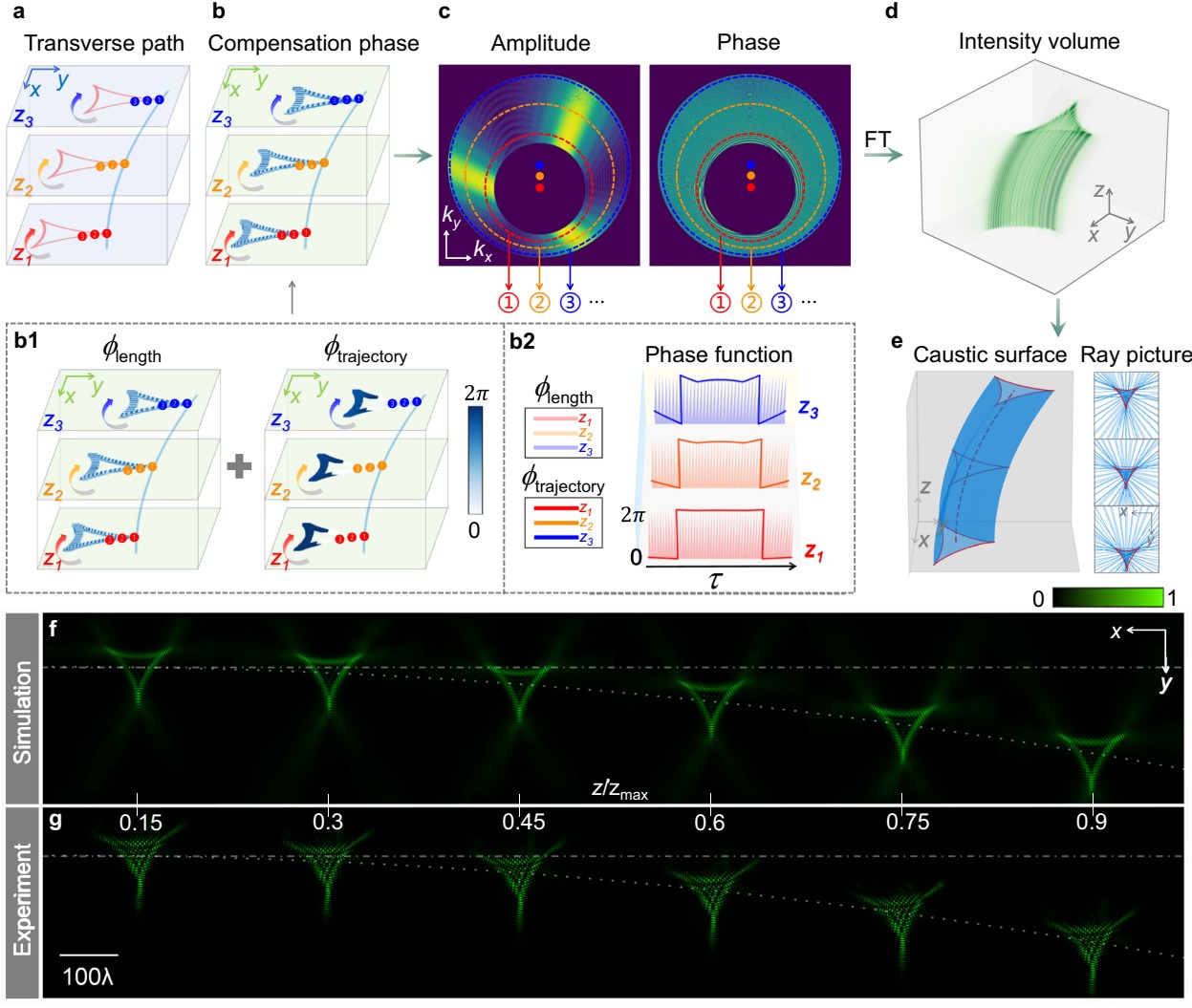

**Fig. 3 | Design and characterization of deltoid-shaped caustic beams with a parabolic trajectory. a** Schematic of the optical caustic paths at transverse planes. **b** Compensation phase with components $\phi_{length}$ and $\phi_{trajectory}$ (**b1**), and their phase functions (**b2**). **c** Amplitude and phase distributions in Fourier space. The dots represent the shifting circle centers. **d** Simulated intensity volume over the propagation range. **e** Surface of the caustics (red) and transverse projection of the rays (blue) at $z = 0$, $z = z_{max}/2$, and $z = z_{max}$, respectively. **f** Simulation results of intensity profiles at propagation distance $z = 0.15 z_{max}$, $z = 0.3z_{max}$, $z = 0.45z_{max}$, $z = 0.6z_{max}$, $z = 0.75z_{max}$, and $z = 0.9z_{max}$. **g** Corresponding experimental results.

To ensure that the interference is constructive during the superposition of fields, it is essential to account for the corresponding compensation phase $\phi$ at each superposed focal point, as illustrated in Fig. 3b. The compensation phase is calculated by Eq. (3), which comprises two subcomponents, $\phi_{length}$ and $\phi_{trajectory}$, depicted in Fig. 3b1. The details of the phases are presented in Fig. 3b2. Notably, $\phi_{length}$ exhibits high-frequency oscillations, while $\phi_{trajectory}$ resembles a square wave distribution. For arbitrary plane $z$, this construction method is the same as described above. Consequently, based on Eq. (4), we can compute the amplitude and phase distribution in Fourier space, as shown in Fig. 3c (see Supplementary Note 3 for detailed phase information). The amplitude and phase information indicated by circles ①, ②, ③ correspond to the propagation distances $z_1$, $z_2$, and $z_3$, respectively. By using angular spectrum transformation, the desired structured light beam in physical space is obtained (see Supplementary Note 11 for the complex-amplitude information in real space at the initial plane). Figure 3d displays the 3D intensity distribution of the deltoid-shaped structured light beam. Figure 3e shows the caustic surface of the light beams, along with the ray pictures (blue) and caustics (red) at $z = 0$, $z = z_{max}/2$, and $z = z_{max}$. Simulated intensity at the detector plane can be seen in Fig. 3f. These results

illustrate that the beam maintains a deltoid intensity profile while bending during propagation. The energy flow rotates around the deltoid shape while propagation forward along the curved trajectory (Supplementary Note 12). Furthermore, the experimental results (Fig. 3g) agree well with the simulation, validating the correctness of our complex-amplitude 3D-printed metasurface design and the high accuracy of fabrication. For an in-depth analysis, the effect of fabrication errors on the caustic beams is provided in Supplementary Note 13. To demonstrate the importance of the compensation phase, we consider four scenarios in Supplementary Note 14: without compensation phase $\phi$, with $\phi_{length}$, with $\phi_{trajectory}$, and with $\phi$. We analyze the complex-amplitude distributions in Fourier space and intensity distributions at plane $z = z_{max}/2$. Evidently, the desired complete caustic pattern is sculptured only when $\phi_{length}$ and $\phi_{trajectory}$ are combined. Using this design method, we customize a Z-shaped caustic that follows a parabolic propagation trajectory (Supplementary Note 15). Further results of the measured intensity distributions of geometric-shaped and letter-shaped caustics at different wavelengths are shown in Supplementary Note 16.

These shape-preserving caustic beams with curved propagation trajectories can be considered as quasi-propagation-invariance beams

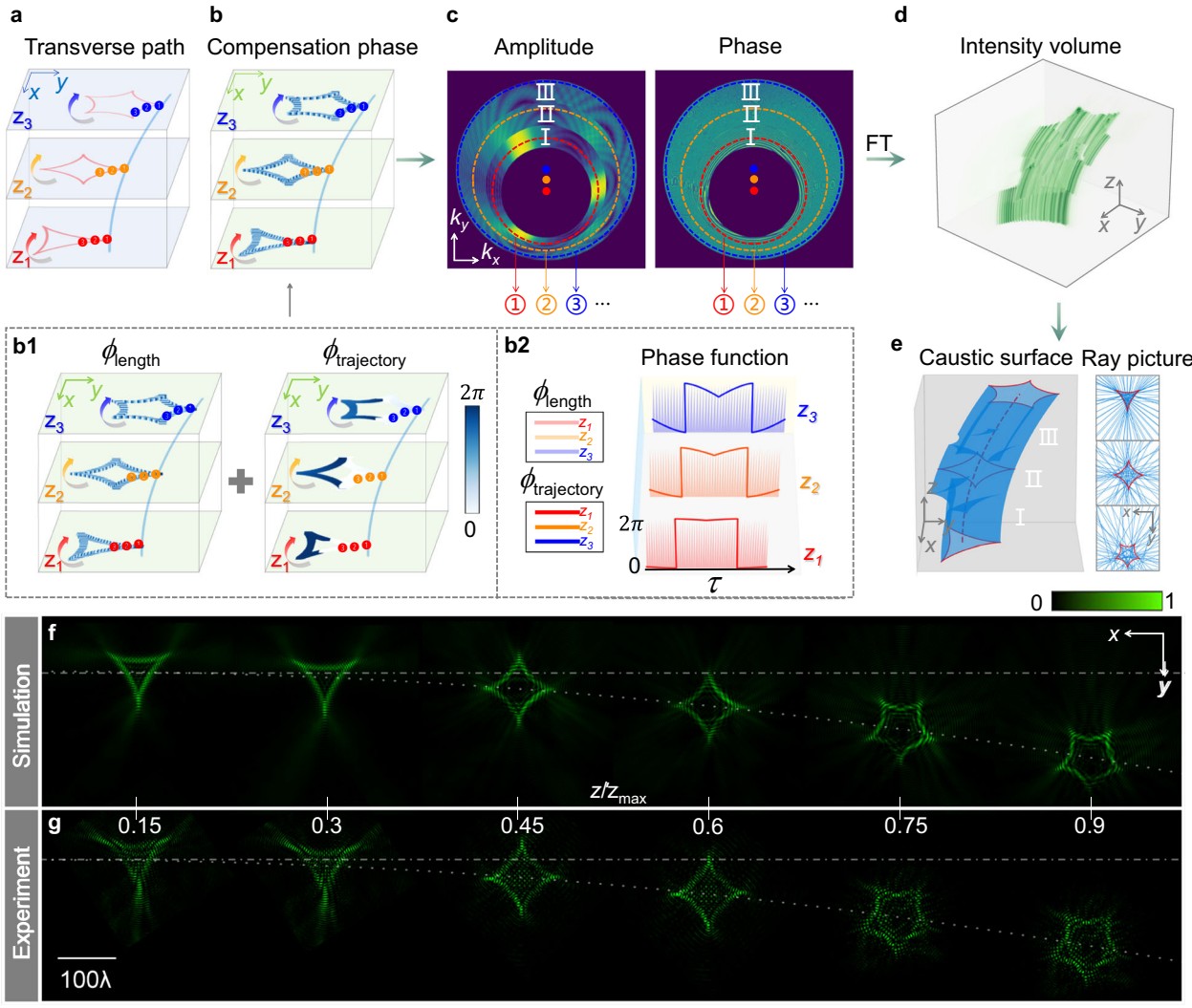

**Fig. 4 | Design and characterization of caustic beams with shapes varying from deltoid to asteroid, and then to hypocycloid with 5 cusps, along a parabolic trajectory. a** Schematic of the optical caustic paths at different transverse planes. **b** Compensation phase, composed of $\phi_{length}$ and $\phi_{trajectory}$ (**b1**), and their respective phase functions (**b2**). **c** Complex-amplitude distributions in Fourier space. The dots are the shifting circle centers. **d** Simulated intensity volume spanning the propagation range. **e** Visualization of the caustic surface and the transverse projections of rays (blue) with desired caustics (red) at the planes $z = 0$, $z = z_{max}/2$, and $z = z_{max}$, respectively. **f** Simulation results of intensity profiles at propagation distances $z = 0.15z_{max}$, $z = 0.3z_{max}$, $z = 0.45z_{max}$, $z = 0.6z_{max}$, $z = 0.75z_{max}$, and $z = 0.9z_{max}$, respectively. **g** Corresponding experimental results.

or diffraction-resistant beams (Supplementary Note 17). Within the preset propagation range, they maintain a relatively stable intensity pattern. Besides, such beams possess a strong self-healing ability, which can be ascribed to the reconstruction by rays at steeper angles that navigate around the obstacles and contribute to the light field far enough behind the disturbance.

The design above is based on the movement of the center of circles and the change in their radii in Fourier space, in order to avoid the overlap of the complex-amplitude information. Specifically, even when radii are constant and some information overlap, caustic shapes can still be effectively generated. This minimal impact on the formation of caustic pattern is due to the small proportion of overlapping information relative to the total complex-amplitude information, provided the design is aptly executed. Some simulation results of the sculpted shapes are showcased in Supplementary Note 18.

## Customizing caustics into morphed patterns with flexible trajectories

Since one caustic shape at plane $z$ is determined by the complex-amplitude information on a ring in Fourier space, multiple caustic shapes can be created at different planes by tailoring the information on corresponding rings. Hence, the intensity profile of the beam dynamically changes along the propagation trajectory.

To demonstrate this concept, we design a morphed light field that not only follows a parabolic trajectory, but also exhibits different caustic shapes in three distinct regions: deltoid caustic when $0 \leq z < z_{max}/3$, astroid caustic when $z_{max}/3 \leq z < 2z_{max}/3$, and hypocycloid with 5 cusps caustic when $2z_{max}/3 \leq z \leq z_{max}$. We use the same trajectory equation and maximum propagation distance as described in the previous sections. The transverse paths of caustic points (red lines) at three propagation distances $z_1$, $z_2$, and $z_3$ are shown in Fig. 4a. Similar to the previous approach, we continuously superpose caustic points such as ❶, ❷, ❸, and so forth, following the direction of the arrows. At each point, we evaluate the corresponding compensation phase $\phi$ (Fig. 4b) to ensure constructive interference. Schematics of the compensation phases ($\phi_{length}$ and $\phi_{trajectory}$) and their respective phase functions are shown in Fig. 4b1, b2. The computed complex-amplitude distributions of the angular spectrum are illustrated in Fig. 4c (see Supplementary Note 3 for detailed phase information), where the circles labeled ①, ②, ③ correspond to the propagation distances $z_1$, $z_2$, and $z_3$, respectively.

The roman numerals I, II, III correspond to three propagation regions. Figure 4d presents the simulated evolving intensity volume throughout the propagation, demonstrating morphed optical caustics. Moreover, Fig. 4e shows the caustic surface, ray pictures (blue) and caustics (red) corresponding to the three propagation regions. The simulation and experimental results of the intensity profiles at different propagation distances are shown in Fig. 4f, g, respectively, with good agreement. Along the parabolic trajectory, the shape of the beam evolves from a deltoid to an astroid, and then to a hypocycloid with 5 cusps. The transverse energy flow rotates around the shapes of deltoid, astriod, and hypocycloid-5, respectively (Supplementary Note 12). The importance of the compensation phase in the construction of this morphed caustic field is shown in Supplementary Note 14. Note that the desired caustic patterns can only be shaped when the entire compensation phase $\phi$ is considered. Further results of the morphed caustic beams with a linear trajectory are shown in Supplementary Note 19.

Thus far, our approach has realized multi-dimensional customization of caustics. The engineered caustics can trace arbitrary contours, creating shapes such as deltoids, astroids, and hypocycloid with 5 cusps. These caustic patterns will propagate along anticipated trajectories. Furthermore, they can morph from one shape to another during propagation.

## Discussion

To summarize, we have demonstrated arbitrary engineering of spatial caustics with morphed intensity patterns along curved trajectories. The engineered caustics are achieved by introducing a compensation phase to ensure constructive interference at every caustic point. The complex-amplitude information of the designed caustic beam in Fourier space is obtained by the method of angular spectrum and the method of stationary phase. This information is transformed into physical space and then encoded into the 3D-printed metasurfaces, which are fabricated by cost-effective, rapid-prototyping, and high-resolution TPL. By varying the height and in-plane rotation angle of each meta-atom (a polymer-based nanofin), we achieve independent and complete control over both the amplitude and phase of the transmitted cross-polarized light. Experimental lensless constructions of caustic structured fields with arbitrary spatial distributions show good agreement with the simulation results. Our approach extends the limited scope of caustic light to arbitrarily customized spatial caustics, with intensities precisely focused along predetermined curves. We anticipate that our results will inspire new ideas for designing more complex propagation scenarios of incoherent caustic fields in future works. Finally, arbitrary caustic engineering can be used in many potential applications. In a variety of complex application scenarios, multi-dimensional control freedom in caustic structured light is required. This attribute endows caustic beams not only with the well-sculpted transverse intensity distribution but also predetermined propagation trajectories. These smart beams can avoid obstructions when the trajectory is partly blocked. In particle manipulation, the curved trajectory and adjustable in-plane intensity distribution facilitate the precise manipulation of microparticles[13,14], enabling diverse trapping patterns and transport paths of small entities such as cells, bacteria, or nanoparticles. It can be employed for applications like cell sorting, drug delivery, and biomedical research. In optical communication, the ability to follow curved trajectories effectively reduces information loss and significantly enhances the robustness of signals[19,49]. In optical imaging, caustic light fields with self-healing properties permit the penetration of complex media such as biological tissues or turbid fluids[17,18]. This property is immensely promising for deep-tissue imaging and medical diagnostic applications. The tailored trajectory and intensity distribution of light beams might enhance the imaging quality, penetrate deep-tissue layers, and offer higher resolution and contrast. In nanofabrication and material processing, these caustic fields enable high-resolution fabrication of complex structures. These capabilities facilitate the direct writing of intricate circuits and photonic devices, advancing the manufacture of next-generation nano-devices with enhanced functionality and performance[15,16,41].

## Methods

### Optical simulation of meta-atoms

The amplitude and initial phase retardation of cross-polarized light beams transmitted through a meta-atom are simulated using the commercial software Lumerical FDTD. The boundary conditions are set to periodic in the $x$ and $y$ directions, and set to perfectly matched layers in the $z$ direction. In the simulation of nanofin, a constant pitch distance of 1.25 μm and a fixed width-to-length ratio of 0.5 are used. The nanofin's height and length are set as sweeping parameters. According to the analytical results based on Jones matrix calculus, the amplitude and initial phase response of a meta-atom are calculated as $A_{out} = |t_l - t_s|$ and $\varphi_{out} = \arctan(t_l - t_s)$, respectively. $t_l$ and $t_s$ are the S-parameters for the polarization along the long axis and short axis, respectively (a detailed derivation is provided in Supplementary Note 20). To design the metasurface with a precise phase distribution, the initial phase introduced by nanofins with different heights is compensated by the PB phase. The initial phase retardation is calculated with respect to a plane above the highest nanofin.

### Fabrication of complex-amplitude 3D-printed metasurfaces

The metasurfaces are fabricated by a Nanoscribe Photonic Professional GT machine, based on TPL within a tightly focused femtosecond laser beam. The polymer metasurface samples are fabricated on indium tin oxide-coated glass substrates using a ×63 Plan-Apochromat objective (NA = 1.40) in the dip-in configuration and IP-L 780 resin (refractive index-1.52 in the visible band). In the fabrication process, ContinuousMode in GalvoScanMode is used, with a scan speed of 7000 μm/s, LaserPower of 90 mW, Slicing (laser movement step in the longitudinal direction) distances of 20 nm, GalvoSettlingTime of 2 ms, and PiezoSettlingTime of 20 ms. Each scanning field is confined to a square unit cell with a size of 100 μm × 100 μm. After the exposure process, the samples are sequentially immersed in propylene glycol monomethyl ether acetate for 15 min, isopropyl alcohol for 5 min (simultaneously exposed by a Dymax BlueWave MX-150 UV LED curing system set at 70% maximum power), and methoxynonafluorobutane for 7 min. The samples are then allowed to dry in ambient air through evaporation.

## Data availability

The data generated in this study are provided in the manuscript and Supplementary Information. The data supporting the findings of this study are available from the corresponding authors upon request.

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

## Acknowledgements

D.M.Z. acknowledges funding support from the Natural Science Foundation of China (NSFC) Grants Nos. 12174338 and 11874321. J.K.W.Y. acknowledges funding support from the National Research Foundation of Singapore (NRFS), under its Competitive Research Programme award NRF-CRP20-2017-0004 and NRF Investigatorship Award NRF-NRFI06-2020-0005. C.-W.Q. acknowledges the financial support of the National Research Foundation, Prime Minister's Office, Singapore under Competitive Research Program Award NRF-CRP22-2019-0006.

## Author contributions

X.Y.Z. and H.T.W. conceived the idea. X.Y.Z. and S.X.L. conducted the theoretical design and numerical simulations of caustic engineering. X.Y.Z., H.T.W., H.W. and C.F.P. contributed to the numerical simulation of nanofin meta-atom. H.T.W. and X.Y.Z. performed the design, numerical simulation, fabrication of 3D-printed metasurfaces. X.Y.Z. and H.T.W. carried out the optical characterization and SEM characterization. X.Y.Z. prepared figures and drafted the manuscript with assistance from H.T.W., J.Y.E.C and H.W. All the authors contributed to the data analysis and manuscript revision. D.M.Z., J.K.W.Y. and C.-W.Q. supervised the project.

## Competing interests

The authors declare no competing interests.
