## [Peer Review File · Nature Communications]

Arbitrary Engineering of Spatial Caustics with 3D-printed MetasurfacesReviewer #1 (Remarks to the Author):

The authors report the use of complex-amplitude metasurfaces to shape caustic fields with curved trajectories in free space. The metasurfaces are fabricated via 3D direct laser writing, in which the height and in-plane rotation angle of 3D nanopillar meta-atom are used to control the amplitude and geometric phase responses of transmitted light, respectively. They also showed the simultaneous generation of multiple different-shaped caustic fields along a curved trajectory. Overall, the results are interesting, and the manuscript is well written. However, the authors should clarify my following points before I can make a recommendation to the Editor.

1. The major concept of this work is to shape caustics in a curved propagation trajectory. However, the authors did not clarify the advantage(s) of doing so. Does this scarify the propagation-invariant feature of the engineered caustics? For instance, Ref. 22 reported arbitrarily shaped caustics with propagation invariance, with the axial size of the non-diffraction beams at least one order of magnitude larger than the transverse beam size.
2. Following my point 1, creating propagation-invariant caustics with self-healing features is highly important for structured light applications. The authors should demonstrate explicitly the propagation-invariant feature of their caustics beams along a curved trajectory and how does it compare to Ref. 22.
3. The authors should explain why the phase response in the Fourier plane is nearly flat. It seems likely that this is an amplitude-only metasurface, how about its diffraction efficiency? Does it really require complex-amplitude modulation?
4. The authors should mention how did they experimentally block the off-center beam at the metasurface plane?

Reviewer #2 (Remarks to the Author):

The application of caustics in structured light faces challenges in tailoring caustic fields with flexible propagation paths and in-plane intensity variations. In this manuscript, the authors introduce the "compensation phase" through 3D-printed metasurfaces to sculpt caustic fields featuring curved trajectories in open space. Furthermore, the in-plane caustic configurations can persist or transform from one structure to another during propagation. The metasurface is facilitated by the rapid and cost-effective two-photon polymerization lithography. This paper could be considered for publication in Nature Communications after the authors answer the following questions:

1. The authors show that the caustic shapes can follow any propagation trajectory achieved by shifting the centers of circular rings in the Fourier space. If the radii of the circular rings remain unchanged and some information overlaps, can the designed caustic shapes still be generated?
2. Usually in order to achieve caustics along a linear trajectory, the centers of circular rings in Fourier space need to be fixed in one position. Altering the information on different rings will lead to mutual interference. It is challenge to create morphed caustics within distinct regions along the linear trajectory. Can the caustic engineering method proposed in this paper achieve the morphed caustic shapes along a linear trajectory?
3. Line 115 should be corrected to $X(z) = 0, Y(z) = 2 \times 10^{-7} z^2$, lying on the plane $x=0$.
4. The period (1250 nm) of nanofin is higher than the working wavelength (532nm), will the high-order diffraction affect the performance of the element? It needs to be addressed.
5. What is the condition to use Eq. (1)? How does the energy flow be controlled along the trajectory if required?
6. Usually the fabricated size of metasurface is small due to the technique difficulty. In this case, please address how the interference pattern can still generate the trajectories designed by ray optics as shown in Fig. 2b?
7. What's the fabrication accuracy of the 3D structure? How will the fabrication error affect the trajectories?
8. The authors claim "We anticipate that our results will inspire new ideas for designing more complex

propagation scenarios of incoherent caustic fields in future works.". Please address more how inherent light could form caustic fields without or with very limited capability of interference.

9. Previous experimental papers also on caustic beam using metasurface should be cited, for example: Optics Letters 45, 551-554 (2020).

Reviewer #3 (Remarks to the Author):

This manuscript focuses on the generation of complex structured light fields using nanoprinted metasurfaces. Specifically, by exploiting the properties of rectangular polymer meta-atoms, the authors have demonstrated that caustic beams with varying properties along the propagation directions can be achieved by using a complex design principle. This design principle is based on a Fourier transform of a caustic beam in k-space and a careful and in-depth analysis of this. Methodologically, the stationary phase method is used. The implementation of the structures is carried out through the use of 3D nanoprinting, obtaining large scale metasurfaces composed of elements with very high aspect ratios. From an experimental standpoint, the validity of the approaches has been demonstrated using a simple custom-made optical setup.

From my perspective, the presented study contains several interesting aspects and I appreciate the overall concept of complex beam generation using 3D nanoprinted metasurfaces. Overall, the manuscript reports unique results related to the generation of structured beams, and I do not see any major technical weaknesses that would fundamentally prevent publication. My main concerns relate to (i) the lack of integration of the results with the existing literature and (ii) the failure to highlight the novelty of the work. At this stage, I cannot judge whether the level of novelty is sufficient for publication in Nature Communications. Therefore, I urge the authors to include a comparison to already published works in the manuscript and use this to highlight the novelty and gain in knowledge. Details can be found under point 1 of my comments below. Overall, I opt for Major Revisions at this stage and will make my decision on the manuscript based on the responses to these comments. I would like to reiterate that my comments are in no way intended to devalue the study, although the results need to be clearly placed in the context of the ongoing research of other groups.

Comments

1. In my opinion, the results are not placed in the context of studies already published in the literature. At this stage, the paper is not suitable to convey to the reader exactly what the benefits of the study presented here are. For example, I am not an expert in this field and therefore cannot judge whether the results are entirely new or whether similar beam shaping experiments have already been carried out with other (maybe simpler) structures. In my opinion, this point is essential for the publication of this study in Nature Communications and needs to be addressed in a further version. I can only agree to publication of this work if a proper comparison is available and a clear degree of novelty is demonstrated by this comparison (ideally by benchmark parameters in a table).

2. At the beginning of the manuscript, the authors discuss various types of metastructures in detail, although the discussion of 3D nano-printed metastructures falls short in my opinion. In particular, the number of references to these structures, which form the central part of the paper, is, in my opinion, too low. I therefore suggest to at least include the following work to give the reader a comprehensive overview about the field: Nat Commun 14, 7222 (2023).

3. It would be very helpful if the authors could include one or more SEM images of a single element into the manuscript. This would allow the reader to accurately assess the precision of the nanoprinting process in the context of the experiments performed here.

4. The aspect described in the previous point is also important, as the large-scale structures shown in Figure 2 show individual elements with rounded surfaces. It would be useful to include in the

manuscript a simulation-based analysis of the influence of these inaccuracies on the optical properties.

5. The experiments shown here are based on the interaction of circularly polarised light with the metastructure. To actually generate the caustic beam, a polarisation unit must be introduced after the metastructure. It would be interesting if the authors could discuss in the manuscript (possibly based on a literature study or simulations) whether it is possible to simplify the beam generation in such a way that a polarisation unit is no longer required. This would also significantly increase the relevance of the study for potential applications.

6. The discussion of the possible applications of the results presented here is very brief and, in my view, not sufficient for publication in Nature Communications. I would therefore ask the authors to discuss the possible application scenarios in much more detail in the discussion section and to clearly demonstrate that their results can be useful in selected applications.

Reply to reviewers and changes made accordingly:

Reviewer 1:

"The authors report the use of complex-amplitude metasurfaces to shape caustic fields with curved trajectories in free space. The metasurfaces are fabricated via 3D direct laser writing, in which the height and in-plane rotation angle of 3D nanopillar meta-atom are used to control the amplitude and geometric phase responses of transmitted light, respectively. They also showed the simultaneous generation of multiple different-shaped caustic fields along a curved trajectory. Overall, the results are interesting, and the manuscript is well written. However, the authors should clarify my following points before I can make a recommendation to the Editor."

We thank the reviewer for the positive evaluation of our research and constructive comments. We provide point-to-point responses below.

Comment 1: The major concept of this work is to shape caustics in a curved propagation trajectory. However, the authors did not clarify the advantage(s) of doing so. Does this scarify the propagation-invariant feature of the engineered caustics? For instance, Ref. 22 reported arbitrarily shaped caustics with propagation invariance, with the axial size of the non-diffraction beams at least one order of magnitude larger than the transverse beam size.

Reply: We agree with the reviewer that we should clarify the advantages of shaping caustics in a curved propagation trajectory.

As we know, caustics are the envelopes of families of rays. The focusing of rays will yield bright edges with high intensities in comparison to the surrounding area. By virtue of the ray structure, the caustics of the beams can be designed as a same shape during the curved propagation trajectory. Hence, the intensity distributions of the caustic beams along the curved propagation trajectory will not be influenced substantially. That is to say, our caustic structured light can maintain a relatively stable intensity pattern. **They can be considered as quasi propagation-invariance beams or diffraction-resistant beams.** The axial non-diffraction distance of our caustic beams can exhibit 2-3 orders of magnitude larger than the transverse dimension. The detailed discussion on the propagation-invariant feature of the engineered caustics can be found in the Reply to Comment 2.

In addition, we would like to point out that, apart from the concept of curved trajectories, our study demonstrates the morphed caustic shapes during propagation. This an important aspect, which has not been achieved in existing literatures. The approaches in Ref. 22 are limited to the linear propagation of a single caustic pattern. Moreover, our research utilizes the advanced two-photon polymerization lithography techniques to fabricate complex-amplitude 3D-printed metasurfaces. So that, it enables both the miniaturization and large-scale manufacturing of optical elements, significantly broadening the potential of arbitrary caustic engineering within the field of nano-optics.

In the revised manuscript, we have added some discussions about the research motivations in the introduction section and possible application scenarios in the discussion section.

...For example, caustic Airy beams are associated with fold catastrophes^{7,8}, Pearcey beams correspond to cusp catastrophes⁹⁻¹¹, and Swallowtail beams represent swallowtail catastrophes¹². Such caustics have found advanced applications in optical trapping^{13,14}, material processing^{15,16}, high-resolution microscopy^{17,18}, and communication technology¹⁹. Nevertheless, these applications face limitations due to a major hurdle in engineering caustic fields with customizable propagation trajectories and in-plane intensity profiles. A leap forward in the multi-dimensional customization of caustics will unlock considerable benefits. For example, these artificial caustic beams can strategically circumvent obstacles, providing an advanced mechanism for the 3D manipulation of particles^{13,14}. This innovation also holds promise in optical communications, where it can minimize signal loss and enhance system stability¹⁹. Additionally, caustic light with precisely defined curvilinear borders facilitates customized high-energy transfer, which is particularly beneficial in nanofabrication^{15,16}. Evolving from natural occurrences to structured light, and now advancing towards advanced engineering, optical caustics are poised for significant breakthroughs, not only in theoretical research and experimental techniques, but also in practical applications (Supplementary Note 1).

...Finally, arbitrary caustic engineering can be used in many potential applications. In a variety of complex application scenarios, multi-dimensional control freedom in caustic structured light is required. This attribute endows caustic beams not only with the well-sculpted transverse intensity distribution but also predetermined propagation trajectories. These smart beams can avoid obstructions when the trajectory is partly blocked. In particle manipulation, the curved trajectory and adjustable in-plane intensity distribution facilitate the precise manipulation of microparticles^{13,14}, enabling diverse trapping patterns and transport paths of small entities such as cells, bacteria, or nanoparticles. It can be employed for applications like cell sorting, drug delivery, and biomedical research. In optical communication, the ability to follow curved trajectories effectively reduces information loss and significantly enhances the robustness of signals¹⁹. In optical imaging, caustic light fields with self-healing properties permit the penetration of complex media such as biological tissues or turbid fluids^{17,18}. This property is immensely promising for deep-tissue imaging and medical diagnostic applications. The tailored trajectory and intensity distribution of light beams might enhance the imaging quality, penetrate deeper tissue layers, and offer higher resolution and contrast. In nanofabrication and material processing, these caustic fields enable high-resolution fabrication of complex structures. These capabilities facilitate the direct writing of intricate circuits and photonic devices, advancing the manufacture of next-generation nano-devices with enhanced functionality and performance^{15,16,40}.

Comment 2: Following my point 1, creating propagation-invariant caustics with self-healing features is highly important for structured light applications. The authors should demonstrate explicitly the propagation-invariant feature of their caustics beams along a curved trajectory and how does it compare to Ref. 22.

Reply: These are good suggestions. The demonstrations of the propagation-invariant and self-healing features have been added to Supplementary Note 17. We have also modified the main text, as follows:

...Further results of the measured intensity distributions of geometric-shaped and letter-shaped caustics at different wavelengths are shown in Supplementary Note 16.

These shape-preserving caustic beams with curved propagation trajectories can be considered as quasi propagation-invariance beams or diffraction-resistant beams (Supplementary Note 17). Within the preset propagation range, they maintain a relatively stable intensity pattern. Besides, such beams possess a strong self-healing ability, which can be ascribed to the reconstruction by rays at steeper angles that navigate around the obstacles and contribute to the light field far enough behind the disturbance.

Supplementary Note 17. The propagation-invariant and self-healing features of caustic beams.

To demonstrate the propagation-invariant feature of the caustic beams, we need to quantify the similarity of the transverse intensity distributions $I_0(x, y)$ at the reference plane with transverse intensity distributions $I_z(x, y)$ at any longitudinal position z . An effective approach is to calculate the normalized cross-correlation function of the intensity distribution patterns²⁴:

$$\gamma(m, n) = \frac{\sum_{x,y} (I_0(x, y) - \bar{I}_0) \cdot (I_z(x-m, y-n) - \bar{I}_z)}{\sqrt{\sum_{x,y} (I_0(x, y) - \bar{I}_0)^2 \sum_{x,y} (I_z(x-m, y-n) - \bar{I}_z)^2}}. \quad (48)$$

\bar{I}_0 and \bar{I}_z are the average values of I_0 and I_z , respectively. (m, n) is the offset when the two intensity matrices are not aligned. If these two matrices are aligned and equal in size, we can set $m = n = 0$. Thus, Eq. (48) is simplified to

$$\gamma = \frac{\sum_{x,y} (I_0(x, y) - \bar{I}_0) \cdot (I_z(x, y) - \bar{I}_z)}{\sqrt{\sum_{x,y} (I_0(x, y) - \bar{I}_0)^2 \sum_{i,j} (I_z(x, y) - \bar{I}_z)^2}} = \frac{cov(I_0, I_z)}{\sigma_{I_0} \sigma_{I_z}}, \quad (49)$$

where $cov(\cdot)$ is the covariance between I_0 and I_z ; σ is the standard deviation.

Here, we calculate z -dependent normalized cross-correlation for four cases of caustic structured light (deltoid, astroid, hypocycloid-5, and line segment) with a

parabolic trajectory, as shown in Fig. S16. The trajectory equation and maximum propagation distance are the same as in Fig. 2 of the main text. The reference plane is chosen as the middle position. Our analysis reveals that within the specified range of propagation distances, the intensity cross-correlation coefficients of the caustic beams predominantly exceed 50%. Further calculations yield average cross-correlation coefficients of these caustic beams with various shapes: 73.8% for deltoid, 63.1% for astroid, 56.0% for hypocycloid-5, and 65.6% for line segment. It is found that all correlation coefficients are greater than 50%, indicating a relatively strong correlation between the fields, and the caustic beam is resistant to diffractive effects.

The field distribution of beams may be perturbed by obstacles or turbulence. Self-healing refers to the process of reconstructing the original structure after such disturbances. Here, we show the self-healing property of the caustic structured light by placing an obstacle in its propagation path. Figure S17 shows the simulation results of blocking a rectangular area (indicated by the white square box in the first column) in the plane $z = 0.5z_{\max}$ of the caustic structured light. One corner of the caustic pattern is obscured. During the propagation process, the caustic structure damaged by a partial obstruction gradually recovers. At the distance of $z = 0.58z_{\max}$, four types of structures are almost completely repaired. In order to describe the degree of self-healing of the beams detailly, we calculated the normalized cross-correlation coefficients between the deltoid-shaped caustic beams blocked by square obstacles with sides $L = 40\lambda$, 60λ , and 80λ , respectively, and the unblocked beams at different propagation distances, together with the intensity distributions at $0.5z_{\max}$ and $0.85z_{\max}$, as shown in Fig. S18.

The white square box indicates the obstructed area. From the quantitative calculation, we can find that the normalized correlation coefficient appears a slight oscillation in the early stage, but when the propagation distance is far enough, it becomes higher and higher and finally tends to a stable value. This suggests that the influence of the obstacle is getting weaker and the lateral light intensity distribution is getting more similar. Moreover, the larger the obstacle size, the smaller the correlation coefficient, the slower the self-healing process, and the longer the recovery distance required. These presented results demonstrate that the caustic beams possess self-healing properties. In the aspect of optical caustics, self-healing after such distortions can be attributed to a reconstruction by rays with a higher inclination that passes the obstacles and contributes to the light field sufficiently far behind the perturbation.

[24] Kaso, A. Computation of the normalized cross-correlation by fast Fourier transform. *PLoS ONE* 13, e0203434(2018).

Fig. S16. Normalized cross-correlations as a demonstration of the invariance of deltoid, astroid, hypocycloid-5, and line segment caustics to obstructions.

Fig. S17. The self-healing of the caustic structured light upon propagation. The blocked area is indicated by the white square box in the first column.

Fig. S18. The normalized cross-correlation coefficients between deltoid-shaped caustic beams blocked by obstacles with different sizes and those not affected by obstacles upon propagation. The blocked area is indicated by the white square box.

Comment 3: The authors should explain why the phase response in the Fourier plane is nearly flat. It seems likely that this is an amplitude-only metasurface, how about its diffraction efficiency? Does it really require complex-amplitude modulation?

Reply: There is abundant phase information on the Fourier plane. To see the details more clearly, we provide magnified images and histograms of the phase distributions in Supplementary Note 3. They correspond to three cases of Fig. 2-4 in the main text. We have also provided an explanation in the revised main text as follows:

...The calculated amplitude and phase distributions in Fourier space are shown in Fig. 2a (see Supplementary Note 3 for detailed phase information).

...Consequently, based on Eq. (4), we can compute the amplitude and phase distribution in Fourier space, as shown in Fig. 3c (see Supplementary Note 3 for detailed phase information).

...The computed complex-amplitude distributions of the angular spectrum are illustrated in Fig. 4c (see Supplementary Note 3 for detailed phase information) ...

Supplementary Note 3. Detailed phase information in Fourier space.

Fig. S2. Phase distributions together with their magnified images and histograms for three cases in the main text.

In our work, the complex-amplitude modulation is indeed needed. It can be seen from the mathematical derivation in the manuscript. For spatial focal curves, the phase distribution is directly given by Eq. (1) in the main text, and the amplitude in the region with phase distribution is unity. For other complex caustic structured light field designs, the phase can be obtained by taking the amplitude angle of Eq. (4) in the main text, and the amplitude is its absolute value.

The diffraction efficiency of our metasurfaces is 15.41%-19.59%, which is calculated as the ratio of the power diffracted into the desired order to the total power incident on the metasurface. After the reviewer's reminder, this information has been supplemented in the main text, as follows:

...As such, the influence of these rounded surface features on the optical performance is marginal. Through an analysis of the ratio between the laser power diffracted into the desired order and the total incident power on the metasurface, we can determine that the diffraction efficiency of our metasurfaces ranges from 15.41% to 19.59%.

Comment 4: The authors should mention how did they experimentally block the off-center beam at the metasurface plane?

Reply: We thank the Reviewer for the comments related to the off-center beam. Here are two types of off-center beams in our work. The first is the caustic structured light beam with a curved propagation trajectory we designed. The second is the higher-order diffractions caused by the metasurface.

For the former, as the propagation distance increases, the beam displaces across the transverse plane. Only one caustic pattern presents at each plane z . Thus, there is no need to block this type of off-center beam at the metasurface plane.

The later are the ± 1 and ± 2 orders diffraction beams in our experiments. However, they do not enter the microscope objective after the metasurface, due to their large diffraction angles ($|\beta_1| \approx 25.2^\circ$ for ± 1 orders, and $|\beta_2| \approx 58.3^\circ$ for ± 2 orders). The detection of the zeroth-order beam will not be affected. Therefore, we do not need to block this type of off-center beam at the metasurface plane. The derivation of higher-order diffractions has been presented in the Supplementary Note 9, and the corresponding explanation has been provided in the main text, as follows:

...then analyzed by another circular polarizer (composed of a quarter waveplate and a linear polarizer with 45° angle). The higher-order diffraction patterns will not enter the microscope objective due to large diffraction angles (see Supplementary Note 9).

Supplementary Note 9. The derivation of higher-order diffractions.

Fig. S7. Cross-sectional schematic of the modeled metasurface.

Figure S7 shows the considered meta-atom, with a nanofin on a glass substrate with refractive index $n_\alpha = 1.5$. The period of the nanofin $P = 1250$ nm. The light wave ($\lambda = 532$ nm) travels from the glass medium to the air with refractive index $n_\beta = 1$.

The incident angle is α , and the refraction angle is β . The higher-order diffraction takes place at points of constructive interference, where the criterion is that the difference in optical path length along the two paths equals an integer number of the vacuum wavelength. This condition is expressed as: $m\lambda = P(n_\beta \sin \beta - n_\alpha \sin \alpha)$ with m being the diffraction order. In our case, the incident angle can be regarded as $\alpha = 0$. Thus, only diffraction orders $m = 0, \pm 1$, and ± 2 occur. The angle of emergence for the first-order diffraction is $|\beta_1| \approx 25.2^\circ$, and for the second-order diffraction is $|\beta_2| \approx 58.3^\circ$.

Reviewer 2:

"The application of caustics in structured light faces challenges in tailoring caustic fields with flexible propagation paths and in-plane intensity variations. In this manuscript, the authors introduce the "compensation phase" through 3D-printed metasurfaces to sculpt caustic fields featuring curved trajectories in open space. Furthermore, the in-plane caustic configurations can persist or transform from one structure to another during propagation. The metasurface is facilitated by the rapid and cost-effective two-photon polymerization lithography. This paper could be considered for publication in Nature Communications after the authors answer the following questions:"

We express our appreciation for the Reviewer's positive feedback and insightful comments. Each question will be addressed in the following.

Comment 1: The authors show that the caustic shapes can follow any propagation trajectory achieved by shifting the centers of circular rings in the Fourier space. If the radii of the circular rings remain unchanged and some information overlaps, can the designed caustic shapes still be generated?

Reply: This is a good question. When the radii of the circular rings remain constant and some information overlap between different rings, as long as the design is appropriate such that the proportion of overlapping information to the total complex-amplitude information is small, the impact on the construction of the caustic pattern is minimal. We have provided simulation results of sculpted shapes under this scenario within Supplementary Note 18 and have accordingly updated the main text as follows:

...contribute to the light field far enough behind the disturbance.

The design above is based on the movement of the center of circles and the change in their radii in Fourier space, in order to avoid the overlap of the complex-amplitude information. Specifically, even when radii are constant and some information overlap, caustic shapes can still be effectively generated. This minimal impact on the formation of caustic pattern is due to the small proportion of overlapping information relative to the total complex-amplitude information, provided the design is aptly executed. Some simulation results of the sculpted shapes are showcased in Supplementary Note 18.

Supplementary Note 18. Intensity profiles under the condition of equal radii of Fourier rings.

Fig. S15. Simulation results of intensity profiles sculpted under the condition of equal radii of Fourier rings: deltoid (a), hypocycloid-6 (b), line (c), Chinese character for “two” (d).

Comment 2: Usually in order to achieve caustics along a linear trajectory, the centers of circular rings in Fourier space need to be fixed in one position. Altering the information on different rings will lead to mutual interference. It is challenge to create morphed caustics within distinct regions along the linear trajectory. Can the caustic engineering method proposed in this paper achieve the morphed caustic shapes along a linear trajectory?

Reply: This is a good question. The caustic engineering method proposed in our manuscript can achieve the morphed caustic shapes along a linear trajectory. Arbitrary caustic engineering is achieved through the combined modulation in both longitudinal and transverse planes. As the reviewer pointed out, if the propagation trajectory is preset as a straight line, the centers of the circles on the Fourier plane will remain fixed, and changing the information on different circles will lead to multiple interference. To circumvent this problem, we first design a curved propagation trajectory, which will cause the movement of the circle centers. Then, we distribute the Fourier information of different caustic patterns across various circles and compensate for the lateral displacement caused by the curved trajectory on the corresponding x - y plane, that is, an additional linear displacement for each caustic shape. In this way, we can realize the morphed caustic patterns along a linear trajectory. We have included this discussion as Supplementary Note 19, and modified the main text, as follows:

...Note that the desired caustic patterns can only be shaped when the entire compensation phase ϕ is considered. Further results of the morphed caustic beams with a linear trajectory are shown in Supplementary Note 19.

Supplementary Note 19. Morphed caustics with a linear trajectory.

Arbitrary caustic engineering is achieved through the combined modulation in both longitudinal and transverse planes. If the propagation trajectory is preset as a straight line, the centers of the circles on the Fourier plane will remain fixed, and changing the information on different circles will lead to multiple interference. To circumvent this problem, we first design a curved propagation trajectory, which will cause the movement of the circle centers. Then, we distribute the Fourier information of different caustic patterns across various circles and compensate for the lateral displacement caused by the curved trajectory on the corresponding x - y plane, that is, an additional linear displacement for each caustic shape. In this way, we can realize the morphed caustic patterns along a linear trajectory. For illustration, we present the simulation results of caustic beams with shapes varying from deltoid to astroid, and then to hypocycloid with 5 cusps, along a linear trajectory, as shown in Fig. S20.

Fig. S20. Propagation dynamics of caustic beams with shapes varying from deltoid to astroid, and then to hypocycloid with 5 cusps, along a linear trajectory.

Comment 3: Line 115 should be corrected to $X(z) = 0$, $Y(z) = 2 \times 10^{-7} z^2$, lying on the plane $x=0$.

Reply: Sorry for the typo. In the revised manuscript, we have corrected this sentence, and doublechecked all the equations.

We present an example of the spatial focal curve with a parabolic trajectory denoted as $X(z) = 0$, $Y(z) = 2 \times 10^{-7} z^2 / \lambda$, lying on the plane $x=0$. The maximum propagation distance...

Comment 4: The period (1250 nm) of nanofin is higher than the working wavelength (532nm), will the higher-order diffraction affect the performance of the element? It needs to be addressed.

Reply: We thank the reviewer for this constructive suggestion. In our study, we consider the meta-atom with a nanofin on a glass substrate ($n_\alpha = 1.5$). The refractive index period of the nanofin $P = 1250$ nm. The light wave ($\lambda = 532$ nm) travels from the glass medium to the air with refractive index $n_\beta = 1$. The incident angle is α , and the refraction angle is β . The higher-order diffraction takes place at points of constructive interference, where the criterion is that the difference in optical path length along the two paths equals an integer number of the vacuum wavelength. This condition is expressed as: $m\lambda = P(n_\beta \sin \beta - n_\alpha \sin \alpha)$ with m being the diffraction order. In our case, the incident angle can be regarded as $\alpha = 0$. Thus, only diffraction orders $m = 0$, ± 1 , and ± 2 occur. The angle of emergence for the first-order diffraction is $|\beta_1| \approx 25.2^\circ$, and for the second-order diffraction is $|\beta_2| \approx 58.3^\circ$.

Therefore, due to the large diffraction angles, the higher-order diffraction patterns separate far from the zero-order pattern after a short propagation distance, and will not affect the zero-order pattern. After the reminder of the reviewer, we realize that it is necessary to clarify this issue. We have included discussion above as Supplementary Note 9, and modified the main text, as follows:

...then analyzed by another circular polarizer (composed of a quarter waveplate and a linear polarizer with 45° angle). **The higher-order diffraction patterns will not enter the microscope objective due to large diffraction angles (see Supplementary Note 9).**

Supplementary Note 9. The derivation of higher-order diffractions.

Fig. S7. Cross-sectional schematic of the modeled metasurface.

Figure S7 shows the considered meta-atom, with a nanofin on a glass substrate with refractive index $n_\alpha = 1.5$. The period of the nanofin $P = 1250$ nm. The light wave ($\lambda = 532$ nm) travels from the glass medium to the air with refractive index $n_\beta = 1$.

The incident angle is α , and the refraction angle is β . The higher-order diffraction takes place at points of constructive interference, where the criterion is that the difference in optical path length along the two paths equals an integer number of the vacuum wavelength. This condition is expressed as: $m\lambda = P(n_\beta \sin \beta - n_\alpha \sin \alpha)$ with m being the diffraction order. In our case, the incident angle can be regarded as $\alpha = 0$. Thus, only diffraction orders $m = 0, \pm 1, \text{ and } \pm 2$ occur. The angle of emergence for the first-order diffraction is $|\beta_1| \approx 25.2^\circ$, and for the second-order diffraction is $|\beta_2| \approx 58.3^\circ$.

Comment 5: What is the condition to use Eq. (1)? How does the energy flow be controlled along the trajectory if required?

Reply: There are two conditions to use Eq. (1). The first one is that the light beam should be paraxial, since we start by employing the paraxial angular spectrum method to analyze the propagation of an optical wave in the Supplementary Note 1. The second one is that the parameter $0 \leq \beta(\tilde{z}) \leq k$, which limits the maximum propagation distance of the caustic point.

Regarding the energy flow distributions of the caustic light fields, we have provided details in Supplementary Note 12, and explain it in the main text, as follows:

...These results illustrate that the beam maintains a deltoid intensity profile while bending during propagation. **The energy flow rotates around the deltoid shape while propagation forward along the curved trajectory (Supplementary Note 12).**

...Along the parabolic trajectory, the shape of the beam evolves from a deltoid to an astroid, and then to a hypocycloid with 5 cusps. **The transverse energy flow rotates around the shapes of deltoid, astriod, and hypocycloid-5, respectively (Supplementary Note 12).**

Supplementary Note 12. The energy flow of the caustic structured light.

The energy flow density vector, also known as the Poynting vector, refers to the amount of energy per unit time flowing through a unit cross section perpendicular to the direction of light propagation^{22,23}. In the paraxial regime, the time average of the energy flow density vector is given by,

$$\mathbf{S} = \mathbf{S}_z + \mathbf{S}_\perp = \frac{1}{2\eta_0} |E|^2 \hat{\mathbf{z}} + \frac{i}{4\eta_0 k} (E \nabla_\perp E^* - E^* \nabla_\perp E), \quad (44)$$

where $\eta_0 = \sqrt{\mu_0 / \varepsilon_0}$ is the impedance of free space; $\hat{\mathbf{z}}$ is the unit vector in the z direction; \mathbf{S}_z and \mathbf{S}_\perp denote the longitudinal and transverse components, respectively. Among them,

$$\mathbf{S}_\perp = \mathbf{S}_x + \mathbf{S}_y \quad (45)$$

with

$$\mathbf{S}_x = \frac{i}{4\eta_0 k} \left(E \frac{\partial}{\partial x} E^* - E^* \frac{\partial}{\partial x} E \right) \quad (46)$$

and

$$\mathbf{S}_y = \frac{i}{4\eta_0 k} \left(E \frac{\partial}{\partial y} E^* - E^* \frac{\partial}{\partial y} E \right). \quad (47)$$

We set $\eta_0 = 377\Omega$, and calculate relative value of the components of the energy flow up light propagation. Fig. S9 shows the energy flow distribution of deltoid-shaped caustic beams corresponding to the case illustrated in Fig. 3 in the main text. In this case, we present the energy flow components at propagation distance $z = 0.5z_{\max}$. The white arrowheads in the first column denote the magnitude and direction of the transverse energy flow. It can be seen that the energy flow rotates around the deltoid shape. The longitudinal component of the energy flow is always positive, and is much larger than the transverse component (about two orders of magnitude). Therefore, the energy flow rotates and moves in the y direction during forward propagation.

Similarly, the energy flow distribution of the morphed caustic beams corresponding to the case illustrated in Fig. 4 of the main text is shown in Fig. S10. Rows 1, 2, and 3 display the transverse and longitudinal components of the energy flow density vector at $z = 0.3z_{\max}$, $z = 0.6z_{\max}$, and $z = 0.9z_{\max}$, respectively. Note that we have shifted the intensity profiles to the middle of the figures for easier observation. In three distinct regions, the transverse energy flow rotates around the shapes of deltoid, astriod, and hypocycloid-5, respectively.

[22] Broky, J., Siviloglou, G. A. Dogariu, A. & Christodoulides, D. N. Self-healing properties of optical Airy beams. *Opt. Express* 16, 12880-12891 (2008).

[23] Gao, X.-Z. et al. Redistributing the energy flow of tightly focused ellipticity-variant vector optical fields. *Photon. Res.* 5, 640-648 (2017).

Fig. S9. The energy flow density vector at $z = 0.5z_{\max}$ for the case illustrated in **Fig. 3** of the main text. The white arrowheads in the first column denote the distribution of the transverse energy flow.

Fig. S10. The energy flow density vector for the case illustrated in Fig. 4 of the main text. Rows 1, 2, and 3 display the energy flow distributions at propagation distance $z = 0.3z_{\max}$, $z = 0.6z_{\max}$, and $z = 0.9z_{\max}$, respectively. The white arrowheads in the first column denote the distribution of the transverse energy flow.

Comment 6: Usually, the fabricated size of metasurface is small due to the technique difficulty. In this case, please address how the interference pattern can still generate the trajectories designed by ray optics as shown in Fig. 2b?

Reply: Thanks for the careful reading of the manuscript by the reviewer. In our research, the fabricated size of the metasurface is two orders of magnitude larger than the wavelength. According to the principles of optics, it's in a regime where both geometric optics and diffraction optics are relevant. Geometric optics simplifies the propagation of light using ray, while diffraction optics describes light using the concept of waves, focusing on the wavefront, phase, and interference. Therefore, the interference patterns can generate the trajectories predicted by ray optics as shown in Fig. 2b. Combining these two theories provides a more comprehensive description of light behavior.

Comment 7: What's the fabrication accuracy of the 3D structure? How will the fabrication error affect the trajectories?

Reply: We thank the Reviewer for raising the questions about the fabrication accuracy and the effects of errors.

Due to the ellipsoid shape of the diffraction-limited focal voxel in 3D laser printing, the rounded surfaces can be observed in our fabricated nanofins. We have established a 3D model based on the single element, and conducted a simulation-based analysis of the influence of rounded corners on the optical properties. Results shows that the discrepancy in both amplitude and phase modulation between nanofins with sharp and rounded edges is small. As such, this rounded effect in the fabricated nanofins exhibits a negligible effect on the optical performance.

In addition, the fabrication errors are caused by machine errors. They are typically regarded as random errors, and their characteristics can be effectively represented and simulated by setting random functions. We have explored the specific impact of different random factors on the caustic light field. The simulation results indicate that as the random factors increase, the disorder of the light field is significantly intensified. However, the random factors introduced by fabrication errors are actually less than 0.2, so the properties of the optical field will not be significantly affected.

We have provided the analysis of fabrication accuracy in Supplementary Note 8, and the effects of errors in Supplementary Note 13. Accordingly, we have revised the main text, as follows:

...More SEM images of nanofins with different heights (3.0 μm to 5.9 μm) are available in Supplementary Note 7. Even though rounded surfaces can be observed in our fabricated nanofins due to the ellipsoid shape of the diffraction-limited focal voxel in 3D laser printing, the discrepancy in both amplitude and phase modulation between nanopillars with sharp and rounded surfaces is small (Supplementary Note 8). As such, the influence of these rounded surface features on the optical performance is marginal.

...validating the correctness of our complex-amplitude 3D-printed metasurface design and the high accuracy of fabrication. For an in-depth analysis, the effect of fabrication errors on the caustic beams is provided in Supplementary Note 13.

Supplementary Note 8. The effect of rounded surfaces of a nanofin on the optical performance.

Fig. S6. Simulation analysis of the effect of rounded surfaces of a nanofin on the optical performance. a Schematic of a nanofin with rounded surfaces. **b, c** Simulation results of normalized amplitude and initial phase for nanofins with sharp and rounded edges.

Supplementary Note 13. The effect of fabrication errors on the caustic beams.

The fabrication error is mainly caused by the machine error, which is a randomly distributed error. Here, we demonstrate the effects of this fabrication error on the caustic structured light. We define the random distribution functions $Random(amplitude)$

and $Random(phase)$ to respectively characterize the amplitude and phase fabrication

errors, and use the random factors a and b to respectively measure the random oscillation degree of the amplitude and phase. Thus, the random amplitude is

$$A_{random} = a \times Random(amplitude) \quad \text{and} \quad \text{the} \quad \text{random} \quad \text{phase} \quad \text{is}$$

$\psi_{random} = b \times \text{Random}(\text{phase})$. The total amplitude is $A_{adjusted} = A_0 + A_{random}$ with A_0 being the initial amplitude, and the total phase is $\psi_{adjusted} = \psi_0 + \psi_{random}$ with ψ_0 being the initial phase. We present two different random factors of deltoid-shaped caustic beams in Fig. S11. It can be found that the larger the random factor, the blurrier the beams and the worse the accuracy.

Fig. S11. The effect of fabrication errors with different degrees on the deltoid-shaped caustic beams.

Comment 8: The authors claim “We anticipate that our results will inspire new ideas for designing more complex propagation scenarios of incoherent caustic fields in future works.”. Please address more how inherent light could form caustic fields without or with very limited capability of interference.

Reply: Coherence and interference are two fundamental concepts in the field of optics. Coherence refers to the predictable phase relationship between waves at different points in space and time. It's divided into temporal coherence and spatial coherence. Temporal coherence refers to the consistency of the phase difference over time at a given point in space. It is closely linked to the light's monochromatic nature. Spatial coherence is about the correlation between the phases of light waves at different points across a wavefront. Interference is the phenomenon where two or more coherent light waves meet in space, resulting in an increase or decrease in light intensity due to their phase differences.

It's commonly assumed that only coherent light is capable of producing interference patterns. However, under certain conditions or with specific techniques, incoherent light can also exhibit interference. The key lies in the concept of coherence being relative. It depends on how the light's coherence length compares to the dimensions of the optical element. According to optical coherence theory, the coherence length refers to the range within which the phase of light waves is strongly correlated, capable of producing clear interference patterns. When the optical element is smaller

than the coherence length of the light source, clear interference patterns can emerge. Thus, a comprehensive understanding of the system as a whole is essential. For example, in the literature [Nat. Nanotechnol. 18, 264], researchers managed to achieve spatio-temporal coherence using an incoherent white light source by downsizing spiral phase plates and pairing them with structural color filters. The realized vortex beams exhibited distinct doughnut-shaped intensity profiles, demonstrating that the coherence of the light source didn't hinder the interference pattern.

As we know, when the size of optical elements is larger than the coherence length, the interference of light passing through these elements indeed diminishes due to reduced coherence. On the other hand, the caustic beams form as a direct outcome of interference, initiated by a predesigned initial condition. That means the interference is at the heart of such beams, and hence we may think that caustic beams must be coherent entities. However, researchers have found that coherence does not diminish the propagation properties of some types of caustic beams (e.g., Airy beams [Optica 2, 886] and Pearcey beams [Opt. Lett. 45, 5496]), provided that the caustic beams are properly designed. By constructing the incoherent beam solely of modes that accelerate along the same trajectory, the incoherence does not hamper the acceleration of the Airy beams. By introducing the degree of coherence function into the light source in the frequency domain, partially coherent Pearcey beams would maintain the inherent properties of autofocusing performance and inversion effect. In these cases, the oscillation of the sidelobe turns smooth, and the intensity distribution concentrates on the mainlobe.

Inspired by these pioneering works, we can also construct the incoherent version of the caustic structured light fields proposed in this manuscript, either through advanced technological innovation or ingenious theoretical design. We plan to further study it in our future work.

Comment 9: Previous experimental papers also on caustic beam using metasurface should be cited, for example: Optics Letters 45, 551-554 (2020).

Reply: We thank the reviewer for bringing the related literature to our attention. In the revised manuscript, this paper about caustic beam using metasurface has been cited in the appropriate places, as follows:

...while avoiding higher-order diffraction channels³³⁻³⁸. **The ability of metasurfaces to manipulate light in tighter spaces can lead to more potential applications in caustic engineering³⁹.** Among them, 3D-printed metasurfaces stand out...

[39] Chen, R., Chen, R.P., Zhou, Y., Chen, W. & Ma, Y. Compact generation of arbitrarily accelerating double caustic beams with orthogonal polarizations using a dielectric metasurface. *Opt. Lett.* 45, 551-554 (2020).

Reviewer 3:

"This manuscript focuses on the generation of complex structured light fields using nanoprinted metasurfaces. Specifically, by exploiting the properties of rectangular polymer meta-atoms, the authors have demonstrated that caustic beams with varying properties along the propagation directions can be achieved by using a complex design principle. This design principle is based on a Fourier transform of a caustic beam in k -space and a careful and in-depth analysis of this. Methodologically, the stationary phase method is used. The implementation of the structures is carried out through the use of 3D nanoprinting, obtaining large scale metasurfaces composed of elements with very high aspect ratios. From an experimental standpoint, the validity of the approaches has been demonstrated using a simple custom-made optical setup.

From my perspective, the presented study contains several interesting aspects and I appreciate the overall concept of complex beam generation using 3D nanoprinted metasurfaces. Overall, the manuscript reports unique results related to the generation of structured beams, and I do not see any major technical weaknesses that would fundamentally prevent publication. My main concerns relate to (i) the lack of integration of the results with the existing literature and (ii) the failure to highlight the novelty of the work. At this stage, I cannot judge whether the level of novelty is sufficient for publication in Nature Communications. Therefore, I urge the authors to include a comparison to already published works in the manuscript and use this to highlight the novelty and gain in knowledge. Details can be found under point 1 of my comments below. Overall, I opt for Major Revisions at this stage and will make my decision on the manuscript based on the responses to these comments. I would like to reiterate that my comments are in no way intended to devalue the study, although the results need to be clearly placed in the context of the ongoing research of other groups."

We thank the Reviewer for thinking our results are interesting and unique, and for appreciating the overall concept of our work. We highly treasure these constructive comments, which will be addressed below:

Comment 1: In my opinion, the results are not placed in the context of studies already published in the literature. At this stage, the paper is not suitable to convey to the reader exactly what the benefits of the study presented here are. For example, I am not an expert in this field and therefore cannot judge whether the results are entirely new or whether similar beam shaping experiments have already been carried out with other (maybe simpler) structures. In my opinion, this point is essential for the publication of this study in Nature Communications and needs to be addressed in a further version. I can only agree to publication of this work if a proper comparison is available and a clear degree of novelty is demonstrated by this comparison (ideally by benchmark parameters in a table).

Reply: We thank the Reviewer for the important comments related to the comparison of the novelty. Due to the significant leaps in the understanding and application of caustics, the development of optical caustics can be divided into three distinct stages.

The first stage, caustics in nature. This initial stage focuses on the observation and study of caustics as they naturally occur. It involves the examination of patterns created by the refraction or reflection of light, such as the rainbow patterns formed with waterdrops or the shimmering light patterns at the bottom of a swimming pool. As the dimensionality of the control parameter space increasing, the caustics have been hierarchically classified seven elementary catastrophes, including four cuspid catastrophes (fold, cusp, swallowtail, butterfly) and three umbilic catastrophes (hyperbolic, elliptic, parabolic). The study of natural caustics lays the foundational understanding of light behavior, paving the way for more structured investigations.

The second stage, caustics in structured light. In the last decade, with the advent of new optical devices such as spatial light modulation, scientists have developed methods to embed elementary types of caustics into structured light beams. These beams exhibit diffraction patterns characterized by corresponding caustics and possess specific propagation trajectories. The diverse propagation properties of caustic structured light, such as self-bending, auto-focusing, and diffraction-free, have been successively reported and attracted widespread attention. Shortly after their discovery, applications for these caustic beams were explored in various fields, including imaging, micro-manipulation, laser beam machining, and communication technologies.

While this approach was groundbreaking, it had several limitations. The main focus is on basic types of caustic beams, their caustic diffraction patterns and propagation trajectories are lack of flexibility. Besides, they are usually realized by spatial light modulator in this phase of development. The experimental setups with large pixel sizes were bulky, hindering easy integration and miniaturization. These drawbacks restrict their broader applications. There's an urgent need for further theoretical breakthroughs and the innovative modulation technologies to overcome these limitations.

Emerging in response to the times, we have arrived at the third stage: caustics in advanced engineering (our work). This growth of multi-dimensional customization of caustics stemmed both from increased theoretical understanding of caustics and from the advances in nanofabrication technology. The engineered caustics can trace any desired profiles, producing a variety of shapes, and will propagate along curved trajectories for the purpose at hand. The employed complex-amplitude 3D-printed metasurfaces, characterized by their ultra-thin profiles and sub-millimeter extensions, offer a compact and efficient solution for generating caustics in fields such as optical micromanipulation, high-resolution microscopy, and nanofabrication. Benefiting from the fast-prototyping and cost-efficiency laser printing technology, these metasurfaces can be produced on a large scale. The improved tunability of the artificial caustics in all facets beyond their natural occurrence is realized, which is poised to bring new revolutions to many domains in the future.

To account for the Reviewer's comment, we have enhanced our manuscript by more explicitly outlining the benefits of our study. Additionally, we have included a comprehensive development and comparison of key researches in the field of optical caustics in Supplementary Note 1, and revised the main text. The modifications are as follows:

...For example, caustic Airy beams are associated with fold catastrophes^{7,8}, Pearcey beams correspond to cusp catastrophes⁹⁻¹¹, and Swallowtail beams represent swallowtail catastrophes¹². Such caustics have found advanced applications in optical trapping^{13,14}, material processing^{15,16}, high-resolution microscopy^{17,18}, and communication technology¹⁹. Nevertheless, these applications face limitations due to a major hurdle in engineering caustic fields with customizable propagation trajectories and in-plane intensity profiles. A leap forward in the multi-dimensional customization of caustics will unlock considerable benefits. For example, these artificial caustic beams can strategically circumvent obstacles, providing an advanced mechanism for the 3D manipulation of particles^{13,14}. This innovation also holds promise in optical communications, where it can minimize signal loss and enhance system stability¹⁹. Additionally, caustic light with precisely defined curvilinear borders facilitates customized high-energy transfer, which is particularly beneficial in nanofabrication^{15,16}. Evolving from natural occurrences to structured light, and now advancing towards advanced engineering, optical caustics are poised for significant breakthroughs, not only in theoretical research and experimental techniques, but also in practical applications (Supplementary Note 1).

Supplementary Note 1. The development of optical caustics.

Fig. S1. Roadmap of the development of optical caustics.

Research Journey	Ref.	Caustic types in transverse plane	Arbitrary propagation trajectory (Yes or No)	Arbitrarily morphed caustics during propagation (Yes or No)	Wavefront modulation	Optical element / minimum resolution	Applications
Caustics in nature	[1]	Fold	×	×	N.A.	Natural objects (e.g. water, glass, crystals)	Art and decoration
	[2]	Cusp	×	×			
	[3-5]	Seven elementary catastrophes	×	×			
	[6]	Fold	×	×			
	[7]	Cusp	×	×			
Caustics in structured light	[8]	Swallowtail and butterfly	×	×	Phase-only	Spatial light modulator / 8 μm	Optical trapping [11] High-resolution microscopy [12] Material processing [13] Communication technology [14]
	[9]	Umbilic	×	×			
	[10]	Arbitrary structures	×	×			
	★ This work	Arbitrary structures	✓	✓			

Tab. S1. Comparison of the researches of optical caustics.

References

- [1] Airy, G. On the intensity of light in the neighbourhood of a caustic. *Trans. Camb. Philos. Soc.* **6**, 379 (1838).
- [2] Pearcey, T. XXXI. The structure of an electromagnetic field in the neighbourhood of a cusp of a caustic. *Lond. Edinb. Phil. Mag.* **37**, 311 (1946).
- [3] Berry, M. V. & Upstill, C. Catastrophe optics: morphologies of caustics and their

- diffraction patterns. *Prog. Opt.* **28**, 257-346 (1980).
- [4] Nye, J. F. *Natural Focusing and Fine Structure of Light* (IOP Publishing, Bristol, 1999).
- [5] Arnold, V. I. & Wassermann, G. S. *Catastrophe Theory* (Springer, Berlin, 2003).
- [6] Siviloglou, G. A., Broky, J., Dogariu, A. & Christodoulides, D. N. Observation of accelerating Airy beams. *Phys. Rev. Lett.* **99**, 213901 (2007).
- [7] Ring, J. D. et al. Auto-focusing and self-healing of Pearcey beams. *Opt. Express* **20**, 18955-18966 (2012).
- [8] Zannotti, A., Diebel, Falko. & Denz, C. Dynamics of the optical swallowtail catastrophe. *Optica* **4**, 1157-1162 (2017).
- [9] Mamsch, C., Zannotti, A. & Denz, C. Embedding umbilic catastrophes in artificially designed caustic beams. *CLEO Europe, EF_4_5* (2017).
- [10] Zannotti, A., Diebel, F. & Denz, C. Dynamics of the optical swallowtail catastrophe. *Optica* **4**, 1157-1162 (2017).
- [11] Baumgartl, J., Mazilu, M. & Dholakia, K. Optically mediated particle clearing using Airy wavepackets. *Nat. Photonics* **2**, 675-678 (2008).
- [12] Vettenburg, T. et al. Light-sheet microscopy using an Airy beam. *Nat. Methods* **11**, 541-544 (2014).
- [13] Mathis, A. et al. Micromachining along a curve: femtosecond laser micromachining of curved profiles in diamond and silicon using accelerating beams. *Appl. Phys. Lett.* **101**, 0711101 (2012).
- [14] Zhang, J. C. et al. A 6G meta-device for 3D varifocal. *Sci. Adv.* **9**, eadf8478 (2023).

Comment 2: At the beginning of the manuscript, the authors discuss various types of metastructures in detail, although the discussion of 3D nano-printed metastructures falls short in my opinion. In particular, the number of references to these structures, which form the central part of the paper, is, in my opinion, too low. I therefore suggest to at least include the following work to give the reader a comprehensive overview about the field: *Nat Commun* **14**, 7222 (2023).

Reply: We thank the reviewer's suggestions, and bring the related literature to our attention. In the revised manuscript, we have expanded our discussion on 3D-printed metastructures and added related references, especially those concerning nanofin structures. The reference recommended by the reviewer has been included, as follows:

...Among them, 3D-printed metasurfaces stand out due to their advantages such as less complex post-lithography processes, large-scale fabrication capability, and the ability to sculpt complex 3D structures⁴⁰⁻⁴⁷. One type of meta-atoms is 3D-printed nanofins^{32,45,46}. These structures function as truncated waveguides with sub-micrometer lateral dimensions and several micrometers height. They offer an additional degree of freedom in height control, resulting in larger real-time delays and a broader working bandwidth compared to traditional plasmonic or dielectric nanoantenna⁴⁷. With this unleashed height degree of freedom, anisotropic nanofins offer an extensive 3D meta-

atom library, allowing the independent and complete polarization, phase or amplitude modulation^{32,45,46}. These polymeric metasurfaces are interfaced with fiber end-faces to perform the dispersion engineering of fiber-optic devices⁴⁵ or transform the output into diverse structured light⁴⁶. In an optical configuration involving crossed circular polarization, the phase of the incident beam is spatially modulated based on the Pancharatnam-Berry (PB) phase, whereas the amplitude is independently controlled by adjusting the height of the nanofins³².

References to 3D-printed nanofins:

[31] Ren, H. et al. Complex-amplitude metasurface-based orbital angular momentum holography in momentum space. *Nat. Nanotechnol.* **15**, 948-955 (2020).

[44] Ren, H. et al. An achromatic metafiber for focusing and imaging across the entire telecommunication range. *Nat. Commun.* **13**, 4183 (2022).

[45] Li, C. et al. Metafiber transforming arbitrarily structured light. *Nat. Commun.* **14**, 7222 (2023).

Comment 3: It would be very helpful if the authors could include one or more SEM images of a single element into the manuscript. This would allow the reader to accurately assess the precision of the nanoprinting process in the context of the experiments performed here.

Reply: We thank the reviewer for raising this point. For a better observation of the precision of the nanoprinting, we have inserted the SEM image of a single structure into the revised Fig. 2, and have described it in the main text. Additionally, we have provided the SEM images of the nanofins with different heights (3.0 μm to 3.7 μm) in Supplementary Note 7. The detailed modifications are as follows:

...3D-printed metasurfaces and the constituent nanofins. For a better assessment of the precision of the nanoprinting, the SEM image of a single nanofin at a 45° observation angle has been inserted. More SEM images of nanofins with different heights (3.0 μm to 5.0 μm) are available in Supplementary Note 7.

Fig. 2 The design principle of arbitrary spatial caustic engineering, the fabrication of complex-amplitude 3D-printed metasurfaces, and the characterization of optical caustics. **a** Initial complex-amplitude distributions of the spatial focal curve in Fourier space. The dots are the circle centers corresponding to different propagation distances. **b** Rays emitted from the circles intersect on a specified focal curve. **c** Visualization of the caustic curve and transverse projections of rays (blue) aligning with the caustics (red) at $z=0$, $z=z_{\max}/2$, and $z=z_{\max}$, respectively. **d** Schematic of one meta-atom. The parameters W , L , H and θ denote the width, length, height and in-plane rotation angle of the polymer nanofin, respectively. LCP and RCP refer to left- and right-handed circular polarized light. The height (H) and in-plane rotation (θ) enable independent control of both the amplitude and phase characteristics of the transmitted light. **e, f** SEM images of the metasurfaces for cases in Fig. 1c, d, together with their magnified images in the top- and oblique-views, as well as an image of a single element. **g** Schematic of the experimental setup for optical caustic characterization. LP, linear polarizer; QWP, quarter wave plate; MO, microscope objective; CMOS, complementary metal oxide semiconductor.

Supplementary Note 7. SEM images of nanofins with different heights.

Fig. S4. SEM images of nanofins with heights varying from $3.0\ \mu\text{m}$ to $5.0\ \mu\text{m}$. All nanofins have uniform transverse dimensions ($W = 400\ \text{nm}$ and $L = 800\ \text{nm}$).

Comment 4: The aspect described in the previous point is also important, as the large-scale structures shown in Figure 2 show individual elements with rounded surfaces. It would be useful to include in the manuscript a simulation-based analysis of the influence of these inaccuracies on the optical properties.

Reply: The ellipsoidal shape of the diffraction-limited focal voxel inherent in 3D laser printing results in the manifestation of rounded edges in the nanofins we have fabricated. To comprehensively understand this phenomenon, we have constructed a 3D model about fabricated elements and conducted the simulation-based analysis to evaluate the effects of these shape discrepancies on optical properties. Results reveal that the differences in amplitude and phase modulation between nanofins with distinctly sharp and those with rounded edges are minimal. Therefore, the presence of rounded edges in our fabricated nanofins exhibits a negligible effect on their overall optical performance.

We have provided the analysis of fabrication accuracy in Supplementary Note 8, and accordingly revised the main text, as follows:

...More SEM images of nanofins with different heights (3.0 μm to 6 μm) are available in Supplementary Note 7. Even though rounded surfaces can be observed in our fabricated nanofins due to the ellipsoid shape of the diffraction-limited focal voxel in 3D laser printing, the discrepancy in both amplitude and phase modulation between nanopillars with sharp and rounded surfaces is small (Supplementary Note 8). As such, the influence of these rounded surface features on the optical performance is marginal.

Supplementary Note 8. The effect of rounded surfaces of a nanofin on the optical performance.

Fig. S6. Simulation analysis of the effect of rounded surfaces of a nanofin on the optical performance. a Schematic of a nanofin with rounded surfaces. b, c Simulation results of normalized amplitude and initial phase for nanofins with sharp and rounded edges.

Comment 5: The experiments shown here are based on the interaction of circularly polarised light with the metastructure. To actually generate the caustic beam, a polarisation unit must be introduced after the metastructure. It would be interesting if the authors could discuss in the manuscript (possibly based on a literature study or simulations) whether it is possible to simplify the beam generation in such a way that a polarisation unit is no longer required. This would also significantly increase the relevance of the study for potential applications.

Reply: We thank the reviewer for raising this suggestion. In the following, we will discuss the feasibility of removing the polarization unit after the metastructure. Our exploration will be divided into two distinct scenarios: the first involves incident light that is circularly polarized, and the second concerns incident light that is linearly polarized.

Let us first discuss the possibility of removing the polarization unit under the condition that the light incident on the metasurface is circularly polarized. We utilize the Jones matrix to describe the amplitude and phase responses of a meta-atom. It is assumed that the incident light, coming from SiO₂ substrate side, carries left-handed circular polarization (LCP) $|L\rangle$:

$$|L\rangle = \frac{1}{2} \begin{bmatrix} 1 \\ i \end{bmatrix}. \quad (\text{S1})$$

The state of light as a function of propagation distance z through the meta-atom can be written as:

$$|\psi(z)\rangle = \Gamma(-\theta)M(z)\Gamma(\theta)|L\rangle, \quad (\text{S2})$$

with

$$M(z) = \begin{bmatrix} A_o(z)e^{i\varphi_o(z)} & 0 \\ 0 & A_e(z)e^{i\varphi_e(z)} \end{bmatrix}, \quad (\text{S3})$$

and a rotation matrix at an angle of θ

$$\Gamma(\theta) = \begin{bmatrix} \cos \theta & -\sin \theta \\ \sin \theta & \cos \theta \end{bmatrix}. \quad (\text{S4})$$

Here, $A_o(z)$, $A_e(z)$, and $\varphi_o(z)$, $\varphi_e(z)$ denote the amplitude and phase coefficients for the polarization along the long axis and short axis of the nanofin, respectively.

The Jones matrix of the meta-atom can be simplified to:

$$|\psi(z)\rangle = \frac{e^{\frac{i\varphi_o(z)+\varphi_e(z)}{2}}}{2\sqrt{2}} \begin{bmatrix} A_o(z)e^{i\varphi_o} (1 + e^{-i2\theta}) + A_e(z)e^{-i\varphi_o} (1 - e^{-i2\theta}) \\ iA_o(z)e^{i\varphi_o} (1 - e^{-i2\theta}) + iA_e(z)e^{-i\varphi_o} (1 + e^{-i2\theta}) \end{bmatrix} \quad (\text{S5})$$

with

$$\varphi_{\square}(z) = \frac{\varphi_o(z) - \varphi_e(z)}{2}. \quad (\text{S6})$$

The right-handed circular polarization (RCP) component of the light after a propagation

distance H (the height of the meta-atom) is represented as the inner product of $|R\rangle$ and $|\psi(H)\rangle$:

$$S = \langle R | \psi(H) \rangle = \frac{1}{2} [A_o(H)e^{i\varphi_o(H)} - A_e(H)e^{i\varphi_e(H)}] e^{-i2\theta}. \quad (S7)$$

If the metastructure is designed such that no additional polarization unit is needed afterward, the polarization conversion efficiency of the meta-atom should ideally approach one hundred percent. To achieve this, three key conditions must be met:

$\varphi_o(H) = 2n\pi$ (n is an arbitrary integer), $\varphi_e(H) - \varphi_o(H) = \pi + 2m\pi$ (m is an arbitrary integer), and $A_o(H) = A_e(H) = 1$. However, the efficiency of amplitude modulation is

contingent upon the conversion efficiency of cross-circular polarization in this configuration. Consequently, achieving complete conversion of cross-circular polarization and simultaneous modulation of amplitude is not feasible.

Next, we evaluate the feasibility of removing the polarization unit when the light incident on the metasurface is linearly polarized. We can reasonably assume that it is

polarized along the x -axis, characterized by a specific Jones vector $\begin{bmatrix} 1 \\ 0 \end{bmatrix}$. The Eq. (S2)

should be rewritten as

$$|\psi(z)\rangle = \Gamma(-\theta)M(z)\Gamma(\theta)\begin{bmatrix} 1 \\ 0 \end{bmatrix}, \quad (S8)$$

Thus, the output electric field is given by¹:

$$E_{out} = \begin{bmatrix} A_o(H)\cos^2(\theta)e^{i\varphi_o} + A_e(H)\sin^2(\theta)e^{i2\varphi_e} \\ \cos(\theta)\sin(\theta)(A_e(H)e^{i2\varphi_e} - A_o(H)e^{i2\varphi_o}) \end{bmatrix}. \quad (S9)$$

The precise amplitude and phase distributions along the x and y directions are attainable. By exploring the 3D meta-atom library, we can identify appropriate geometric parameters that ensure E_x and E_y exhibit identical amplitude and phase¹. Therefore, in this configuration, the simultaneous modulation of both amplitude and phase of the metasurface can be realized without the need for a polarization unit.

It's worth mentioning that the Capasso's group has reported a dual matrix holography technique, which allows encoding amplitude and phase information onto a phase-only metasurface^{2,3}. The polarization unit after the metasurface can be removed in the configuration, but a 4-f system is required for filtering purposes.

[1] Li, C. et al. Metafiber transforming arbitrarily structured light. *Nat. Commun.* **14**, 7222 (2023).

[2] Dorrah, A.H., Rubin, N.A., Tamagnone, M., Zaidi, A. & Capasso, F. Structuring total angular momentum of light along the propagation direction with polarization-controlled meta-optics. *Nat. Commun.* **12**, 6249 (2021).

[3] Dorrah, A. H., Rubin, N. A., Zaidi, A., Tamagnone, M. & Capasso, F. Metasurface optics for on-demand polarization transformations along the optical path. *Nat. Photonics* **15**, 287–296 (2021).

Comment 6: The discussion of the possible applications of the results presented here is very brief and, in my view, not sufficient for publication in Nature Communications. I would therefore ask the authors to discuss the possible application scenarios in much more detail in the discussion section and to clearly demonstrate that their results can be useful in selected applications.

Reply: We thank the reviewer for raising this suggestion. We have added some discussions about the possible application scenarios in the revised manuscript, as follows:

...Finally, arbitrary caustic engineering can be used in many potential applications. In a variety of complex application scenarios, multi-dimensional control freedom in caustic structured light is required. This attribute endows caustic beams not only with the well-sculpted transverse intensity distribution but also predetermined propagation trajectories. These smart beams can avoid obstructions when the trajectory is partly blocked. In particle manipulation, the curved trajectory and adjustable in-plane intensity distribution facilitate the precise manipulation of microparticles^{13,14}, enabling diverse trapping patterns and transport paths of small entities such as cells, bacteria, or nanoparticles. It can be employed for applications like cell sorting, drug delivery, and biomedical research. In optical communication, the ability to follow curved trajectories effectively reduces information loss and significantly enhances the robustness of signals¹⁹. In optical imaging, caustic light fields with self-healing properties permit the penetration of complex media such as biological tissues or turbid fluids^{17,18}. This property is immensely promising for deep-tissue imaging and medical diagnostic applications. The tailored trajectory and intensity distribution of light beams might enhance the imaging quality, penetrate deeper tissue layers, and offer higher resolution and contrast. In nanofabrication and material processing, these caustic fields enable high-resolution fabrication of complex structures. These capabilities facilitate the direct writing of intricate circuits and photonic devices, advancing the manufacture of next-generation nano-devices with enhanced functionality and performance^{15,16,40}.

We thank again the reviewers for their critical and enlightening reviews that help us improve our manuscript. We hope all the concerns from the reviewers have been addressed, and the manuscript fulfill the requirements for publication.

Reviewer #1 (Remarks to the Author):

I would like to thank the authors for taking care of all the comments very carefully. I have read the whole response letter, and believe this revision is suitable for publication in Nature Communications, which now contains enhanced motivation and highlights of its innovation on caustic fields and 3D laser printing technology.

Reviewer #2 (Remarks to the Author):

The authors have answered my questions and in my view, the manuscript could be accepted for publication at the current version.

Reviewer #3 (Remarks to the Author):

From my point of view, the editors have done a great job in improving the manuscript, which is now ready for publication.